



# Signs of reduced biospheric activity with progressing global warming: evidence from long-term records of atmospheric CO2 mixing ratios in Central-Eastern Europe

Łukasz Chmura[1,2], Michał Gałkowski[1,3], Piotr Sekuła[1,2], Mirosław Zimnoch[1], Jarosław Nęcki[1], Jakub Bartyzel[1], Damian Zięba[1], Kazimierz Różański[1], Wojciech Wołkowicz[4] , Laszlo Haszpra[5]

[1]AGH University of Science and Technology, Faculty of Physics and Applied Computer Science, Krakow, Poland,
[2]Institute of Meteorology and Water Management, National Research Institute, Krakow, Poland
[3]Max Planck Institute for Biogeochemistry, Department of Biogeochemical Systems, Jena, Germany
[4]Polish Geological Institute, National Research Institute, Warsaw, Poland
[5]Research Centre for Astronomy and Earth Sciences, Hungarian Academy of Sciences, Sopron, Hungary

*Correspondence to*: Łukasz Chmura (chmura@fis.agh.edu.pl)

**Abstract.** The recent rise of temperatures across the globe, mainly attributed to the raising anthropogenic emissions of greenhouse gases, is predicted to have an increased impact on ecosystems over the next century and beyond. One of the manifestations of this anthropogenic global warming will be the increased occurrence of prolonged droughts in the temperate climate zones, specifically in Northern America and Europe. Drought events that took place in Europe in 2003 and 2010 are known to have led to significant reduction of carbon dioxide sink, due to simultaneous occurrence of water stress limiting the photosynthetic activity and increase of respiration under higher temperatures. In the current study we present the evidence of an increased impact of droughts on the annual cycle of carbon dioxide over Central-Eastern Europe, based on long-term observations (1995-2018) of mixing ratios conducted at two continental sites: Kasprowy Wierch mountain station (KAS, Southern Poland) and Hegyhatsal tall tower (HUN, Hungary). Analyses of the smoothed, detrended annual cycles from both sites reveal a gradual reduction of annual amplitudes towards lower magnitudes, with simultaneous reductions of annual maxima (KAS: -0.13 ± 0.05 ppm/yr, HUN: -0.08 ± 0.12 ppm/yr) and increases of minima (KAS: 0.09 ± 0.04 ppm/yr, HUN: 0.08 ± 0.08 ppm/yr). By comparing the area of influence of both stations (established by analyses of footprints calculated with Hysplit Lagrangian model) to the regions of drought extent (established by analysing the temperature and soil moisture anomalies), we attribute the observed rising trend of annual minima to the increased frequency of large-scale drought events that reduce mean summer assimilation rates over Central-Eastern Europe. This conclusion is further corroborated by comparison to the biogenic fluxes calculated by the regional inversion system CarbonTracker-EU, albeit the statistical uncertainty is non-negligible ($CO_2$ biogenic flux over Europe Transcomm region is equal to 0.03 ± 0.03 PgC/yr). On the other hand, reduction of anthropogenic $CO_2$ emissions (-0.07 ± 0.02 PgC/yr over Europe) seem to at least partially explain the trend of reduced winter maxima of $CO_2$ at the observation sites.



## 1 Introduction

Reliable predictions of future climate change require application of numerical climate models, which in turn rely on high-
quality observations of atmospheric concentrations of greenhouse gases (GHG) supplied by measurement sites located in
different parts of the world. Monitoring of greenhouse gas concentrations over Europe is conducted through ICOS observation
network (https://www.icos-cp.eu/) and NOAA/ESRL/GMD CCGG cooperative air sampling network
(https://www.esrl.noaa.gov). Central-Eastern Europe is poorly represented in both networks. In NOAA CMDL network there
is currently only one station located in Hegyhatsal, Hungary (Major et al., 2018). In present structure of ICOS network, stations
are missing east of 16°E except of Swedish (Norunda, Svartberget) and Finish (ATM-UTO, Hyytiala, Pallas, Puijo) sites,
representing mainly Scandinavia.

Availability of dense GHG observation networks allows scientists to analyse both long-term trends and anomalies in carbon
balance as well as response of the biosphere to changes of climate observed during the last decades. An example of such study,
made on regional scale, is an investigation of the influence of high air temperatures induced by global warming on reductions
in agricultural productivity in US (Schauberger et al., 2017). Another example, illustrating assessment of climate extremes
(droughts) on vegetation growth for temperate Northern Hemisphere was published by Wu et al. (2018). Similar analyses for
the European region were published by Schewe et al. (2019) and Ciais et al. (2005). The first article points to underestimation
of climate extremes by global models using the 2003 European heat wave event as an example. The second one studies the
reduction in primary productivity of European biosphere during the same event.
This lack of proper representation of Central-Eastern Europe in present GHG observation networks is partly compensated by
Kasprowy Wierch high-altitude greenhouse gas monitoring station located in the High Tatras mountain range of southern
Poland, in operation since 1994. Long-term time series of $CO_2$ and $CH_4$ available for this site, in combination with relevant
data from other stations, especially the Mace Head baseline station (monitoring greenhouse gas characteristics of maritime air
masses entering Europe), enabled to study the impact of continental sources and sinks of GHGs on the observed atmospheric
levels of those gases in the interior of the European continent.

The presented study was focused on seasonal variability of $CO_2$ mixing ratios observed at Kasprowy Wierch and Hegyhatsal
stations, both located in Central-Eastern Europe. Long-term trends of winter concentration maxima and summer concentration
minima of $CO_2$, as well as departures from those trends associated with occurrence of climate extremes (droughts, heat waves)
observed in the years 2003, 2010, 2012, 2014, and 2015 were studied in some detail.

## 2 Site descriptions

### 2.1 Kasprowy Wierch (KAS)

Kasprowy Wierch GHG monitoring station (49°14' N, 19°59' E, 1989 m a.s.l., 400 m above the tree line) is located in the
meteorological observatory situated on top of Kasprowy Wierch peak located in the High Tatras, at the intersection of three



main valleys (Bystra, Gąsienicowa, and Cicha), at the border between Poland and Slovakia. Climate of the area is a typical alpine temperate one, with large diurnal and seasonal variations of temperature, high precipitation rates, frequent changes of atmospheric pressure, and strong winds. One of the important phenomena affecting circulation of the local atmosphere is a frequent occurrence of temperature inversions. The inversion layer inhibits vertical exchange of heat, water vapour, and other constituents of the atmosphere. Local surface winds are controlled by the morphology of the surrounding area (anabatic and katabatic wind events are frequently observed). Wind statistics available for the station indicate prevalence of SW and NE winds, caused by Gąsienicowa and Cicha valleys, channelling the air flow in these directions (Chmura, 2010). However, when long-distance transport of air masses is considered, westerly winds are the dominant feature of air circulation in the lower troposphere. Since Kasprowy Wierch is located within the transition zone between free troposphere and planetary boundary layer and is relatively free of local influences, it is considered a regional background site for trace gas measurements in the lower atmosphere over Central-Eastern Europe.

The observatory building is heated by electricity. Approximately 30 m below the observatory building another structure, housing a cable car station, is located. This building also uses only electricity for heating purposes. Thus, there are no significant sources of anthropogenic GHG emissions in the immediate vicinity of the station, with the exception of snow groomers cars maintaining ski pistes during wintertime. The observatory is a WMO first class synoptic station conducting regular observations of a wide array of meteorological parameters (wind speed and direction, air temperature and pressure, humidity, cloudiness, cloud types, visibility, precipitation, meteorological phenomena, and others). Analytical equipment for GHG measurements is located in one of the rooms of the observatory building. The analysed air is sucked through heated line with the inlet located approximately 5 m above the ground level.

## 2.2 Mace Head (MHD)

Mace Head Research Station is located on the west coast of Ireland (53°20'N, 9°45'W, about 100 m from the Atlantic shore). The station is classified as a global background station within the WMO-GAW network. The site is exposed to the North Atlantic Ocean (clean air sector, 180° through west to 300°). Galway city (population of approximately 65000 inhabitants) is the nearest major conurbation, approximately 90 km to the east of Mace Head. The hilly area around Mace Head is wet and boggy with a lot of exposed rock and vegetation, which consists mainly of grasses and sedges. The facilities at the site consist of three laboratory buildings, two aluminum walk-up towers (20 m and 10 m), and a converted 20 ft cargo container office (https://www.esrl.noaa.gov/gmd/dv/site/; Dlugokencky et al., 2018). The meteorological records show that, on average, over 60% of the air masses arrive at the station via the clean sector. The climate is mild and moist, being dominated by maritime air masses (http://macehead.org/; Jennings et al., 2003; Vardag et al., 2014).

## 2.3 Hegyhatsal (HUN)

The Hegyhatsal tower (46°57'N, 16°39'E, 248 m a.s.l.) belongs to the European network of tall tower sites, which monitor all important greenhouse gases. The tower is surrounded by agricultural fields (mostly crops and fodder of annually changing





types) and forest patches. The measurements are carried out at the level of 96 m above the ground. In the vicinity of the tower there are only small villages, between 100 to 400 inhabitants, in the radius of 10 km around the site. The nearest of them is Hegyhatsal (170 inhabitants), situated approximately 1 km from the tower. No notable industrial activity is present in the area. Local roads have mostly low levels of traffic. One of the few main roads of the region, which on average carries 3600 vehicles

per day, passes approximately 400 m southwest of the tower. Measurements of $CO_2$ mixing ratio profiles, temperature, humidity and wind profiles began in September 1994. The Hegyhatsal tower is also a NOAA/CMDL global air sampling network site and part of the Global Atmosphere Watch programme (https://gaw.kishou.go.jp/search/station; Haszpra et al. 2010; https://www.esrl.noaa.gov/gmd/dv/site/; Dlugokencky et al. 2018).

## 3  Methods

### 3.1 Measurement techniques

Regular observations of $CO_2$ mixing ratios at Kasprowy Wierch station began in 1994. From September 1994 until June 1996 weekly composite samples of air were collected and $CO_2$ mixing ratios were analyzed at the Institute of Environmental Physics, University of Heidelberg, Germany. In 1996, automated gas chromatograph (Agilent HP5890) equipped with Flame Ionization Detector (FID) was installed and quasi-continuous analyses have been carried out ever since (Nęcki et al., 2003; Chmura et

al., 2008; Nęcki et al., 2013). In January 2015 a laser CRDS spectrometer (Picarro, L2201-i) has been installed at the station. Carbon dioxide mixing ratios and its isotopic composition ($\delta^{13}C$-$CO_2$) are now recorded with a frequency of approximately 0.5 Hz. All $CO_2$ mixing ratio measurements are calibrated and are traceable to the international primary scales (WMO-$CO_2$-X2007). The station participated in several inter-laboratory comparisons conducted in the framework of EU Projects as a part of Quality Assurance/Quality Control (QA/QC) program implemented at the station (CUCUMBER 2016).

Both Hegyhatsal and Mace Head are also measuring $CO_2$ using the state of the art techniques and report mixing ratios directly calibrated against WMO-CO2-X2007 scales. Information on their specific instrument set and calibration techniques can be found in Haszpra et al. 2010 and Dlugokencky et al., 2018. The data for both stations were retrieved from the GLOBALVIEW-CO2 NOAA ESRL Carbon Cycle Cooperative Global Air Sampling Network (http://www.esrl.noaa.gov/gmd/dv/data/ – Dlugokencky et al., 2018).

### 3.2 Footprints

Assessment of the area of influence (footprints) for Kasprowy Wierch and Hegyhatsal stations has been carried out for the periods of interests, with the aim of providing information on the distribution of regional $CO_2$ sources affecting atmospheric mixing ratios observed at both stations. This was achieved with the use of a Hybrid Single-Particle Lagrangian Integrated Trajectory model (Hysplit 4, revision 983 – Stein et al., 2015). The model was run in the back-trajectory mode, i.e. retracing

the track of particles that arrived at Kasprowy Wierch and Hegyhatsal stations, thus providing information on the origins of the air masses arriving at the measurement sites. Each trajectory was derived from 96-hours long backward simulations.



Simulations performed for this study were driven by NCEP Reanalysis meteorological data, provided by the NOAA/OAR/ESRL PSD, Boulder, Colorado, USA (https://www.esrl.noaa.gov/psd/), pre-processed into the Hysplit-readable format by Air Resources Laboratory, NOAA (https://ready.arl.noaa.gov/HYSPLIT.php). Particles were released hourly from

points located 1989 m a.s.l. for Kasprowy Wierch and 248 m a.s.l. for Hegyhatsal.

From the hourly releases of particles, an aggregated gridded spatial maps of area of influences were calculated at 0.5° x 0.5° resolution from individual back-trajectories, using the tools available in the Hysplit modelling suite, and then smoothed by focal averaging method with a constant 5 cells x 5 cells weighting field. All trajectory points below 10 km altitude were included in the spatial gridding algorithm. The analyses performed in the context of this study follow an approach similar to

Jeelani et al., 2018.

### 3.3 Surface air temperature and soil humidity

In order to calculate annual air temperature and soil humidity anomalies for summer and winter seasons, ERA-Interim gridded data, providing surface monthly mean values on approximately 80 km spatial resolution, were used (Dee at al., 2011). For temperature, summer anomaly (June, July and August) was defined as a residual from long-term trend calculated over the

period between 1980 and 2018. Similar calculations for winter season (December, January, February) were performed against the reference period from December 1979 to February 2018. Calculations outlined above were made separately for each grid point in the European region.

Soil humidity anomalies were derived as departures from the mean values representing summer and winter volumetric soil water content in the uppermost soil layer (depth from 0 to 7 cm) and calculated for the same reference period as in the case of

temperature (December 1979 to February 2018).

### 3.4 Biogenic and anthropogenic fluxes

Long-term variations in the intensity of the biospheric sink and fossil-fuel related $CO_2$ sources on the European continent can be derived also from carbon cycle models. For the purpose of this study, biospheric and fossil-fuel $CO_2$ fluxes for the European TransCom region (Gurney et al., 2000) for the period 2000-2017 were used. The data is available from CarbonTracker Europe

CTE2016 modelling framework, provided by Wageningen University (Wageningen, NL, http://www.carbontracker.eu; for details see van der Laan-Luijkx et al., 2017). For simplicity, only the optimized biospheric fluxes and pre-assigned fossil-fuel emissions used in the inversion system will be presented here. Emissions from biomass burning and oceans were negligible in the context of the presented study.





## 4 Results and discussion

### 4.1 $CO_2$ mixing ratio records

The record of $CO_2$ mixing ratios available for Kasprowy Wierch station is shown in Fig. 1a. It covers the period from September 1994 to December 2018. The data points shown in Fig. 1a are daily averages of $CO_2$ mixing ratios (green circles in the background) calculated on the basis of quasi-continuous measurements, after applying appropriate data selection procedure aimed at selecting only those data points which are representative for free-atmosphere conditions at the station, not influenced

by nearby sources (Necki et al., 2003, Chmura, 2010). The smoothed record of $CO_2$ mixing ratios (heavy green line) was calculated using the procedure recommended by NOAA (CCGvu 4.40 – Thoning et al., 1989). The same procedure was used to calculate long-term trend line. The trend line served as a basis to calculate growth rates of the $CO_2$ signal; from 1995 to 2018 the $CO_2$ mixing ratios recorded at Kasprowy Wierch increased from 361 ppm in 1994 to 409.5 ppm in 2018, an increase of 48.5 ppm, i.e. 13.4%.

The long-term trend curve of $CO_2$ mixing ratios recorded at Kasprowy Wierch station was compared in Fig. 1b with analogous trend curves calculated for Mace Head and Hegyhatsal stations. Whereas the distance between Kasprowy Wierch and Hegyhatsal stations is only approximately 350 km, they differ greatly in terms of their characteristics: Kasprowy Wierch is a high-mountain station whereas Hegyhatsal is a typical lowland site with much higher impact of the local biosphere on the recorded $CO_2$ mixing ratios. It is apparent from Fig. 1b, that mean growth rates of $CO_2$ mixing ratios at all three sites are

similar (2.02 ppm·year$^{-1}$ for Kasprowy Wierch, 2.07 ppm·year$^{-1}$ for Mace Head, and 2.13 ppm·year$^{-1}$ for Hegyhatsal).

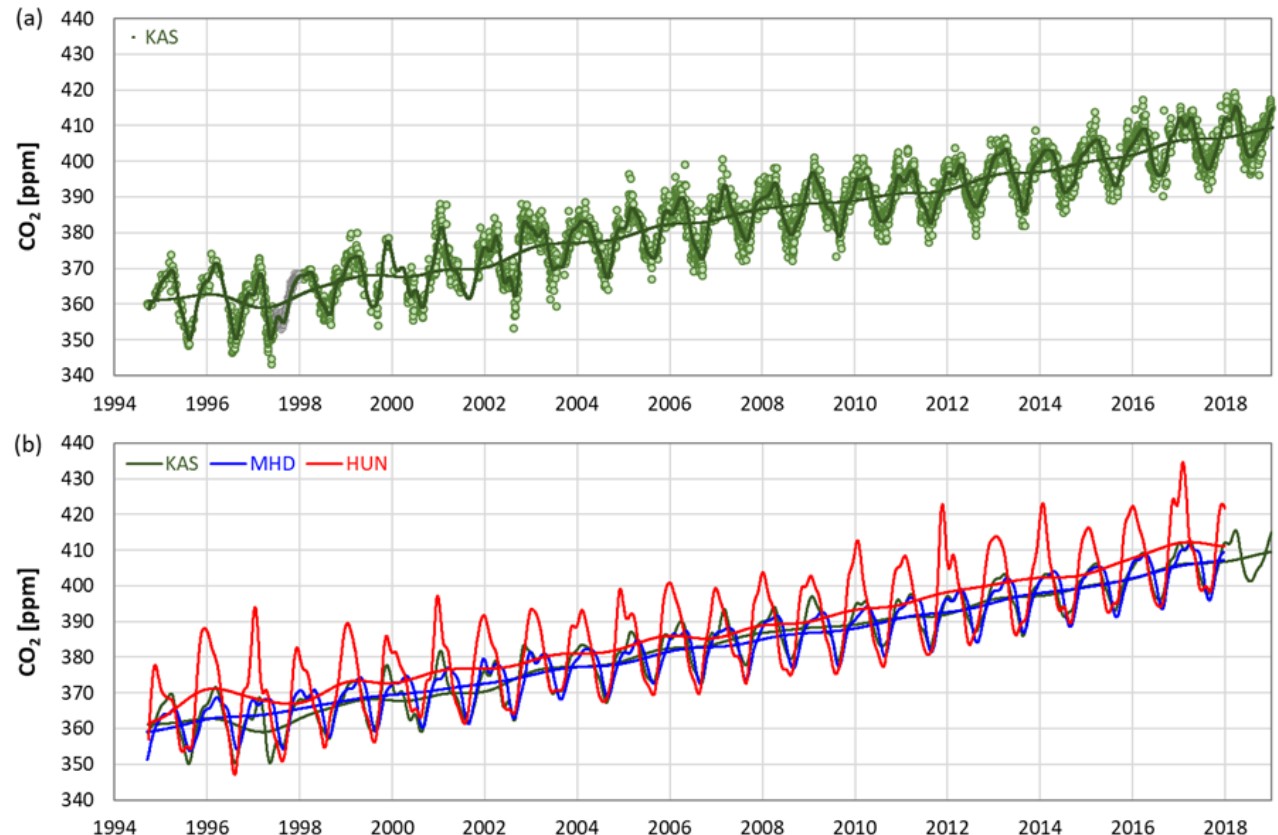

**Figure 1: (a) - Atmospheric CO₂ mixing ratios recorded at Kasprowy Wierch from 1994 to 2018. Green circles (daily averages) represent regional CO₂ signal, free of local influences, calculated from filtered raw measurement data (Necki et al., 2003, Chmura, 2010). Due to instrument malfunction, data between July and December 1997 have been gap-filled with a mean annual cycle calculated from period 1996 – 2018, added to a long-term linear trend. Solid lines - smoothed CO₂ mixing ratios recorded at the station and the long-term trend line of those ratios, respectively (see text). (b) - Smoothed CO₂ mixing ratios and their long-term trend lines calculated for Kasprowy Wierch (KAS, in green), Mace Head (MHD, in blue) and Hegyhatsal (HUN, in red) stations.**

## 4.2 Seasonality of CO₂ mixing ratios recorded in Central-Eastern Europe

The seasonal component of smoothed CO₂ mixing ratio records presented in Fig. 1b can be obtained by subtracting the long-term trend from smoothed CO₂ records. The resulting detrended seasonal CO₂ signals for Kasprowy Wierch, Mace Head, and Hegyhatsal are shown in Fig. 2 as functions of time. The annual CO₂ cycle observed at each station is controlled by the biospheric activity on the European continent and the seasonality of anthropogenic emissions of this gas. Generally higher anthropogenic CO₂ emissions occur over the continent during winter, when, due to lower ambient temperatures, fossil fuel heating sources are in operation. In contrast, low temperatures and low amount of sunlight greatly reduce photosynthetic activity of the biosphere. On the other hand, summers are characterized by lower anthropogenic emissions due to lower overall heating power demand and high biospheric uptake of CO₂. The overall effect of these processes can be seen in the detrended CO₂ records shown in Fig. 2. These provide the basis for calculating peak-to-peak amplitudes of the CO₂ signals.





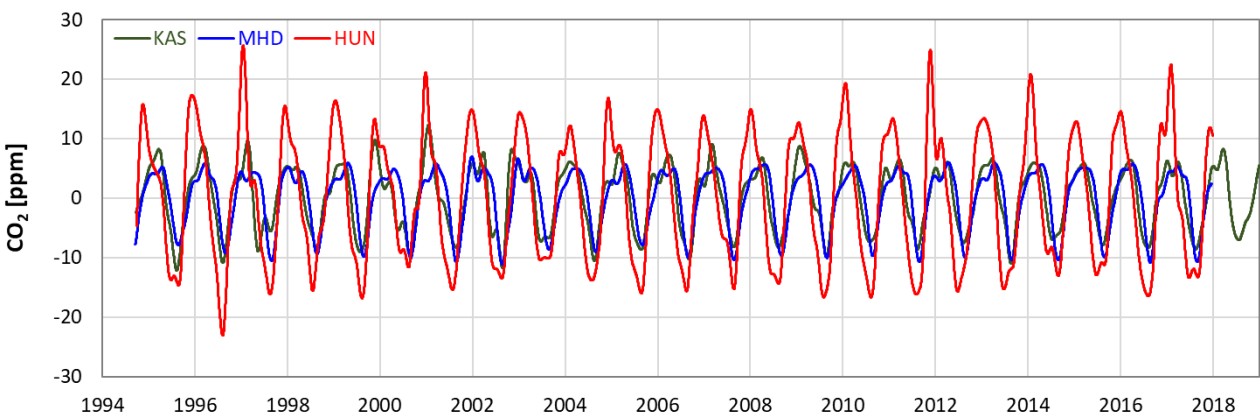

**Figure 2: Temporal evolution of detrended CO₂ mixing ratio records available for Kasprowy Wierch (KAS), Mace Head (MHD), and Hegyhatsal (HUN) stations. The curves were calculated using CCGvu 4.40 routine (Thoning et al., 1989), by subtracting the long-term trends from smoothed CO₂ records.**

Temporal evolution of peak-to-peak amplitudes calculated for Mace Head, Kasprowy Wierch, and Hegyhatsal stations is shown in Fig. 3. It is apparent that although the amplitudes recorded at Hegyhatsal are almost twice as high as those recorded at Kasprowy Wierch, they reveal similar downward trend, albeit in case of Hegyhatsal the uncertainty is high. Amplitude of the seasonal $CO_2$ cycle at Kasprowy Wierch decreases with the rate of $0.23 \pm 0.06$ ppm·year$^{-1}$, to be compared with $0.15 \pm 0.15$ ppm·year$^{-1}$ observed at Hegyhatsal station. In contrast, the Mace Head station representing boundary conditions for the European continent does not reveal any discernible long-term trend in peak-to-peak amplitudes of the recorded $CO_2$ signal (slope of the best fit line $0.05 \pm 0.04$ ppm·year$^{-1}$).


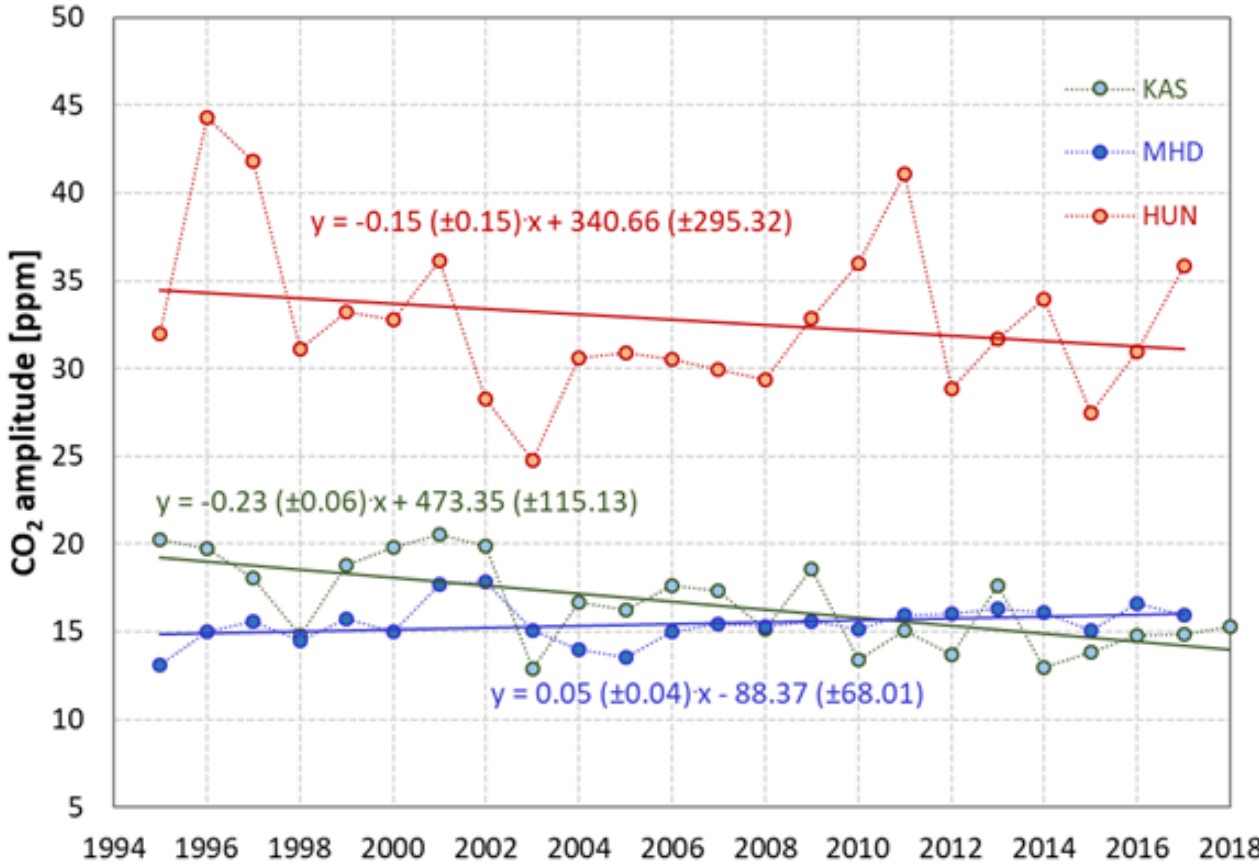

**Figure 3: Temporal evolution of peak-to-peak amplitudes of seasonal variations of $CO_2$ mixing ratios recorded at Kasprowy Wierch (KAS, in green), Mace Head (MHD, in blue), and Hegyhatsal (HUN, in red) stations. For each dataset best fit linear regression line is shown.**

Reduction of peak-to-peak amplitude of seasonal $CO_2$ mixing ratio cycle observed at Kasprowy Wierch and Hegyhatsal (Fig. 3) may result from two effects: (i) higher mixing ratios of $CO_2$ recorded during peaks of summer seasons, and/or (ii) lower $CO_2$ mixing ratios recorded during winter periods. Figure 4 shows peak-to-peak amplitudes of seasonal $CO_2$ cycle recorded at Kasprowy Wierch, compared with the maximum (Fig. 4a) and minimum (Fig. 4b) values of $CO_2$ mixing ratios recorded at this station in the given year. Minimum and maximum $CO_2$ values were calculated based on smoothed and detrended $CO_2$ mixing ratio records. It is apparent from Fig. 4 that the decrease of peak-to-peak amplitudes of seasonal $CO_2$ cycle at Kasprowy Wierch

occurring with the rate of $-0.23 \pm 0.06$ ppm·year$^{-1}$ is caused by an increase of summer minimum of $CO_2$ mixing ratio ($0.09 \pm 0.04$ ppm·year$^{-1}$) and a simultaneous decrease of winter maximum of this ratio proceeding with the rate of $0.13 \pm 0.05$ ppm·year$^{-1}$.



**Figure 4: Comparison of peak-to-peak amplitude of the seasonal CO₂ cycle recorded at Kasprowy Wierch station during the period 1995–2018, with the maximum (a) and the minimum (b) values of CO₂ mixing ratios recorded in the given year. The peak-to-peak amplitude and the minimum and maximum CO₂ values were calculated based on smoothed and detrended CO₂ mixing ratio record obtained with the aid of CCGvu 4.40 routine (Thoning et al., 1989).**

Figure 5 shows analogous comparisons of peak-to-peak amplitudes of the seasonal CO₂ cycle with the maximum (Fig. 5a) and

the minimum (Fig. 5b) values of CO₂ mixing ratios recorded at the Hegyhatsal station, Hungary. It is clear from Fig. 5 that

apparent reduction of peak-to-peak amplitudes of CO₂ seasonal cycle at this station during the period 1995-2017 (0.15 ± 0.15





ppm year$^{-1}$) stems equally from the reduction of winter maxima (-0.075 ± 0.115 ppm·year$^{-1}$) and the increase of summer minima (0.078 ± 0.076 ppm·year$^{-1}$), in the same way as at Kasprowy Wierch station.




**Figure 5:** Comparison of peak-to-peak amplitude of the seasonal CO$_2$ cycle recorded at Hegyhatsal station during the period 1995–2017, with the maximum (a) and the minimum (b) values of CO$_2$ mixing ratios recorded in the given year. The peak-to-peak amplitude and the minimum and maximum CO$_2$ values were calculated based on smoothed and detrended CO$_2$ mixing ratio record obtained with the aid of CCGvu 4.40 routine (Thoning et al., 1989).


The data presented in Figs. 4 and 5 suggest that, over the past 25 years, the biospheric pump removing $CO_2$ from the European atmosphere during summer months shows signs of weakening, and that anthropogenic winter $CO_2$ emissions from the continent are on decline. A combination of both these effects leads to the observed reduction of peak-to-peak amplitude of the $CO_2$ mixing ratio records available for the stations located in Central-Eastern Europe.

To test our hypothesis, we compared the observed $CO_2$ mixing ratio signals with the regional CarbonTracker-EU product (cf. Methods section) presented in Fig. 6 which shows temporal variability of biospheric $CO_2$ and fossil-fuel fluxes for the European continent for the period 2000-2017. The modelled biospheric and fossil-fuel $CO_2$ fluxes over the European continent for the past 18 years were compared with the minimum (Fig. 7a) and maximum (Fig. 7b) values of $CO_2$ mixing ratios recorded at this station in the given year. The minimum and maximum $CO_2$ values were calculated based on smoothed and detrended

$CO_2$ mixing ratio records. The comparison was made for the common period of data availability. It is apparent from the data presented in Fig. 7 that the inverse model corroborates our conclusions, both with respects of decreasing European fossil-fuel flux (the rate of $0.07 \pm 0.02$ Pg $CO_2$·year$^{-1}$) as well as growing net $CO_2$ flux of the continental biosphere (the rate of $0.03 \pm 0.03$ Pg $CO_2$·year$^{-1}$).

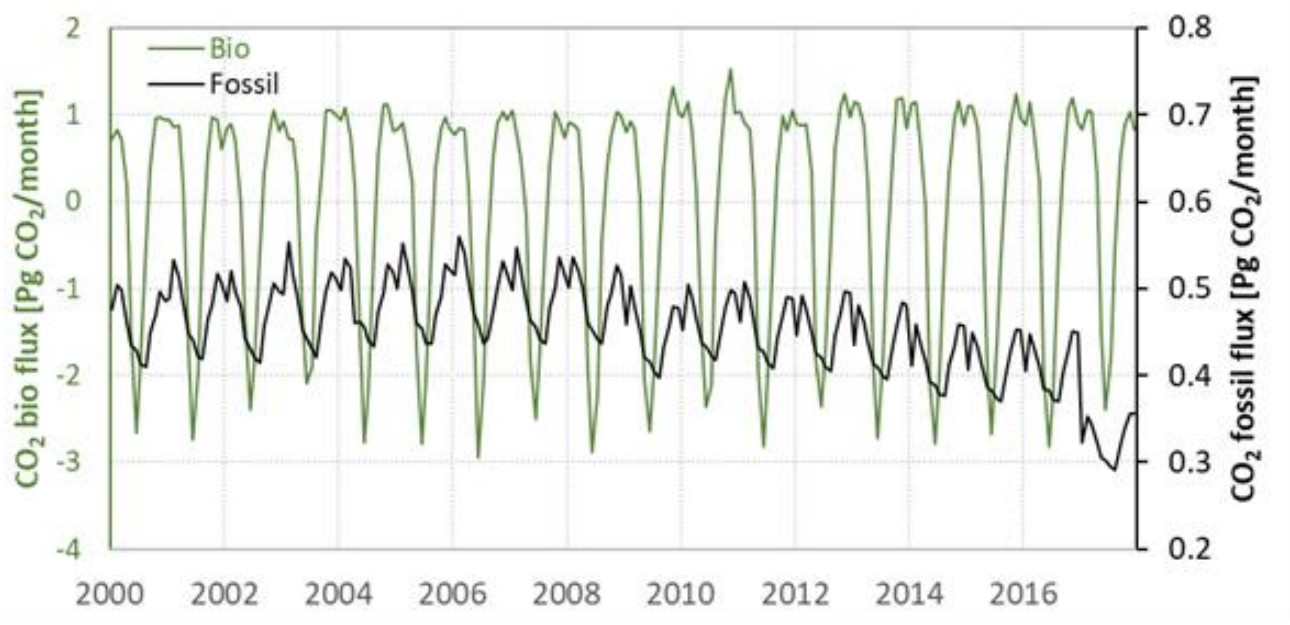


**Figure 6: Temporal variability of the biospheric $CO_2$ (in green) and fossil-fuel (in black) fluxes for the European continent for the period 2000-2017, available from CarbonTracker Europe CTE2016 (Wageningen University, NL, http://www.carbontracker.eu).**





Figure 7: Comparison of $CO_2$ minima (a) and maxima (b) recorded at Kasprowy Wierch with modelled annual biospheric (a) and fossil-fuel (b) fluxes for the European continent, derived from CarbonTracker Europe CTE2016 (van der Laan-Luijkx et al., 2017).

**4.3 Anomalously low peak-to-peak amplitudes of seasonal $CO_2$ cycles at Kasprowy Wierch and Hegyhatsal stations**

As discussed above, low peak-to-peak amplitudes of the seasonal $CO_2$ cycle recorded in a given year at stations located in Central-Eastern Europe indicate weak biospheric $CO_2$ sink during summer and reduced fossil-fuel flux of $CO_2$ into the regional atmosphere during winter, both considered within the area of influence (footprint) of the given station. Weak biospheric sink during summer may result from heat waves and/or reduced availability of moisture in the soil profile. As can be seen in Fig. 4a, there were five years in the data record available for Kasprowy Wierch, with peak-to-peak amplitudes of the seasonal $CO_2$


cycle smaller than 14 ppm: 2003, 2010, 2012, 2014, 2015. The years 2003, 2012 and 2015 also stand out as years with lowest peak-to-peak amplitude in the $CO_2$ record of the Hegyhatsal station (Fig. 3).

Drastic reduction of the biospheric downward $CO_2$ flux over the European continent observed in 2003 was discussed by Ciais et al. (2005), who found that it was mainly caused by heat wave that hit western Europe in the summer of 2003. Weak biospheric pump during summer 2003 caused substantial reduction of peak-to-peak amplitude of the seasonal $CO_2$ cycle over Europe in this year. At Kasprowy Wierch, the $CO_2$ amplitude dropped from 19.9 ppm in 2002 to 12.9 ppm in 2003, and subsequently recovered to 16.7 ppm in 2004. At Hegyhatsal, the observed changes in peak-to-peak amplitudes were smaller:

28.3, 24.8, and 30.6 ppm, respectively. Those changes in peak-to-peak amplitudes of the seasonal $CO_2$ signal can be compared with the spatio-temporal distribution of surface air temperature and soil moisture anomalies, shown in Figs. 8 and 9. The 2003 heat wave is reflected in positive anomaly of surface air temperature for this particular year over western and southern Europe (Fig. 8); in the same year, the soil moisture deficit was recorded over France, Germany, and Balkans (Fig. 9).

Lower impact of 2003 heat wave on peak-to-peak amplitudes of $CO_2$ seasonal signal recorded at Hegyhatsal station, when

compared to Kasprowy Wierch, can be understood by examination of the area of influence (footprint) maps shown in Fig. 10, calculated for summer months (June-August) of 2003. First, due to much higher elevation, the footprint area of Kasprowy Wierch is significantly larger than that of Hegyhatsal station. Secondly, whereas Kasprowy Wierch was receiving air masses arriving predominantly from west and north-west direction covering major part of western Europe mostly affected by heat wave, footprint of Hegyhatsal station was biased towards north-west and northern direction, less affected by this heat wave.

Relatively low values of peak-to-peak amplitudes of the seasonal $CO_2$ cycle were recorded at Kasprowy Wierch station also in 2010, 2012, 2014, and 2015 (13.4 ppm, 13.7 ppm, 13.0 ppm, and 13.9 ppm, respectively). In particular, during summer 2010 the European continent experienced even larger heat wave than in summer 2003, with the mainly affected areas this time located in eastern part of the continent. The heat wave was centered in the southern part of Russian Federation (middle Wolga catchment) and was accompanied by reduced soil moisture availability in this region and large-scale forest fires (Hauser et al.,

2010, Barriopedro et al., 2011). The fact that Kasprowy Wierch $CO_2$ record was affected by the 2010 heat wave whereas Hegyhatsal record apparently not (cf. Figs 3 and 4), can be understood by careful analysis of the footprint maps calculated for both stations for summer months of 2010 (Fig. 10) and the extension of temperature anomaly for this summer shown in Fig. 8. It is evident that the area of influence calculated for Hegyhatsal station does not reach the region affected by the heat wave. In contrast, the area of influence calculated for Kasprowy Wierch extends well into this region. It should be also noted that

summer 2010 was wet in Hungary (cf. Fig. 9), thus enhancing the biospheric activity in the footprint area of the station and deepening the summer minimum of $CO_2$ concentration recorded at this site.

The summers of 2012 and 2015, which are characterized by low peak-to-peak amplitudes of $CO_2$ seasonal cycle at both stations, were moderately warm in Europe (cf. Fig. 8). However, as seen in Fig. 9, they stand out as exceptionally dry years, particularly in southern (2012) and central (2015) Europe. This was probably the reason for weaker biospheric activity and

reduced amplitudes of seasonal cycle of $CO_2$ recorded at Kasprowy Wierch and Hegyhatsal.


**Figure 8: Surface air temperature anomalies for summer season in Europe (months of June, July, and August) during the period 1995-2018, calculated with respect to the reference period 1980-2018, after removing the long-term trend (cf. Methods section).**






**Figure 9: Anomalies of water content in the uppermost soil layer (depth interval 0-7 cm) calculated for summer season in Europe (months of June, July, and August), with respect to the reference period 1980-2018 (cf. Methods section).**





**Figure 10: Footprint maps for Kasprowy Wierch (upper panel) and Hegyhatsal (lower panel) stations calculated for summer months (June, July, August) of 2003 (left-hand panels) and 2010 (right-hand panels). Individual 96-hours backward trajectories were calculated using Hysplit4 at a hourly frequency, aggregated and smoothed to provide the area of influence maps for the discussed period.**

Decreasing amplitudes of atmospheric carbon dioxide seasonal cycle over Central-Eastern Europe, visible in $CO_2$ mixing ratio records available for Kasprowy Wierch and Hegyhatsal stations, have its roots not only in increasingly warmer and dryer summers but also in warmer winters. Figure 11 presents anomalies of mean surface air temperature over the European continent during winter months (December, January, February) calculated with respect to the reference period (1980-2018), after





removing the long-term trend. Warmer winters are associated with reduced fossil-fuel $CO_2$ emissions (less fuel used for heating), leading to reduction of maximum concentrations of this gas in the atmosphere during heating season (December to

February). Good examples of this effect are the years 2008 and 2014, when low peak-to-peak amplitudes of $CO_2$ seasonal cycle were recorded at Kasprowy Wierch (15.2 ppm and 13.0 ppm, respectively). It happened despite the fact that summers were rather cold and wet (Figs. 8 and 9). These two years were characterized by exceptionally mild winters in most parts of central and northern Europe. Average temperatures in December, January, and February exceeded long-term average by as much as 6ºC (Fig. 11).





**Figure 11: Surface air temperature anomalies for winter season in Europe (months of December, January, February) during the period 1995-2018, calculated with respect to the reference period 1980-2018, after removing the long-term trend (cf. Methods section).**




## 5 Conclusions

Biosphere responds to rising levels of atmospheric $CO_2$ and associated global warming in various ways. On one hand, an increase of atmospheric mixing ratios of $CO_2$ increases the rate of photosynthesis by plants (fertilization effect), thus leading to enhanced uptake of $CO_2$ from the atmosphere (e.g. Zhu et al., 2016). On the other hand, elevated temperatures associated

with global warming (heat waves) and/or reduced availability of soil moisture (droughts) slow down the photosynthetic activity of plants resulting in reduced biospheric uptake atmospheric $CO_2$ (e.g. Ciais et al., 2003, Hauser et al., 2016). Long-term, high-quality observations of atmospheric mixing ratios of $CO_2$ provide a powerful tool to trace those subtle, climatically induced changes in the carbon balance over the continents in prolonged time periods.

Analysis of long-term records of atmospheric mixing ratios of carbon dioxide available for two continental sites located in

Central-Eastern Europe (Kasprowy Wierch, Poland, and Hegyhatsal, Hungary) revealed that during the observation period discussed here (1995-2018) the peak-to-peak amplitudes of the seasonal $CO_2$ cycle at those sites show a decreasing trend. This apparent reduction of seasonal $CO_2$ cycle stems most likely from two effects: (i) an increase of summer minima of $CO_2$ concentration due to reduced biospheric uptake of atmospheric carbon dioxide during summer, and (ii) a decrease of winter maxima of $CO_2$ concentration due to reduced consumption of fossil fuels for heating purposes during mild winters. Whereas

reduced biospheric uptake is clearly visible only during exceptionally warm and/or dry summers, reduction of winter maxima induced by progressing warming plays a dominant role in the observed decreasing trend of seasonal $CO_2$ cycle over Central-Eastern Europe, particularly during the past decade.

In-depth analysis of reasons for anomalously low amplitudes of the seasonal $CO_2$ cycle observed from time to time at both stations reveals that they were climatically induced i.e. they were caused by exceptionally warm and/or dry summers or

exceptionally mild winters. Careful analysis of the area of influence for each station helped to disentangle various factors controlling temporal variability of seasonal $CO_2$ cycles recorded at those sites. From this perspective, high-altitude mountain station Kasprowy Wierch appears to be better suited for tracing subtle, long-term changes of carbon cycle over the European continent than low-altitude Hegyhatsal station because of larger footprint of Kasprowy Wierch and lack of strong biospheric signals in immediate vicinity of the station.

**Acknowledgements**

The authors wish to thank numerous individuals and institutions which supported the measurements of trace gas composition of the atmosphere at Mace Head, Kasprowy Wierch, and Hegyhatsal stations over the past 23 years. The current work was partially supported by the Polish Ministry of Science and Higher Education (projects no. 16.16.220.842 B02 and 211350/E-356/SPUB/2016/1-1). Hungarian National Research, Development and Innovation Office supports the measurements at

Hegyhátsál under the contract OTKA K129118.





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
