# Peer review of "Signs of reduced biospheric activity with progressing global warming: evidence from long-term records of atmospheric CO2 mixing ratios in Central-Eastern Europe"

_Atmospheric Chemistry and Physics, 2019_

## Referee Comment (RC1) · Anonymous Referee #1 · 4 Dec 2019

**General comments**

The study examines the changes in the atmospheric $CO_2$ seasonal cycle amplitude (SCA) at two Central-Eastern Europe stations, Kasprowy Wierch (KAS) and Hegyhatsal (HUN) since 1994. More specifically, the authors analyse seasonal minimum and maximum, and their trends and associations to the changes in climate conditions. Understanding the seasonal cycle of atmospheric $CO_2$ is important as it is affected by the seasonal cycle of both natural and anthropogenic fluxes, which can change due to climate change and mitigation actions.

They found that the atmospheric $CO_2$ SCAs at those sites have decreased since 1994, due to an increase in summer minimum and a decrease in winter maximum. These trends correlate well with $CO_2$ fluxes estimated from a model (CTE2016), where the European biospheric fluxes have increased and the European anthropogenic emissions have decreased. These results suggest that changes in biospheric activities and a reduction in fossil fuel emissions could be examined from the measurements. It also warns potential future changes; plants may be suffering from sever summers even more, and fertilization effect may be weakening, and thus there is an urgent need in faster reduction of anthropogenic emissions.

The two selected sites have high quality long-term records, which are suitable for examining such long trends. It is a slight concern that they only use two sites to conclude about regional $CO_2$ fluxes, but they have examined the trajectories of those sites, which support the appropriateness of those two sites for this study.

The manuscript is generally written carefully with considerations, but I would like to address a few points below, which the authors could further discuss more in depth.

- Major concern is that the authors conclude that the changes in the SCA have been driven only by increase in biospheric fluxes in summer and fossil fuel emissions in winter. Other causes, such as an extension of plants' glowing seasons, effects of warm winter to biospheric fluxes or warm summer effects in anthropogenic emissions (see below) have not been discussed. The authors have examined exceptional years, but are there any sign of increasing trends of such years (e.g. mild winters and warm/dry summers)? In addition, if the increase in biospheric fluxes is a major cause in increase of summer minimum atmospheric $CO_2$, it is due to reduction of photosynthesis or increase in respiration? How about feedbacks, e.g. fertilization effects (only one sentence about fertilization in the conclusion)?

- The authors have concluded that the trends and events of the changes in the SCA is associated with climate conditions, but did not discuss in detail the effects of anthropogenic actions, such as land use change and policy or economical conditions. Also, references to show that fossil fuel emissions have decreased is missing (CTE and thus its prior is from a model). In addition, the seasonal cycle of anthropogenic emissions could have changed as well – warmer winter, and more sever summer could cause shift in fossil fuel emissions from winter heating to summer cooling.

- The evaluation with flux sources are done by comparing flux estimates from a model. As footprints of summer and winter are shown to dominate from Central-Western Europe, how much about Eastern Europe can you say? The CTE results are from whole Europe, so how do

you incooperate your arguments considering the trajectories? To be more precise, source contributions should be calculated by running trajectories or transport model for each sources separately. This would require extra work, but increases the value of this study significantly.

- Using results from only one model in evaluation should be done with caution. Any models have their own features, and results may be biased. In addition, if those studied sites were assimilated in the CTE inversion (i.e. has influence on the flux results), the evaluation is not independent of the measurements. Please specify and justify the choice, and if possible, use other sources, such as results from Global Carbon Project for biospheric fluxes and UNFCCC reports for anthropogenic emissions as additional source of evaluation.

**Specific comments**
P2 L42-43: The advantage of GHG observations are useful not only for studying last decades, but also present and future. It is true that "currently available" observations can only see the past, but findings from those studies can be used for future predictions as well.

P2 L48-19: What was their main conclusion?

P2 L50-53: Please rephrase. It is not clear whether you wish to say that KAS is useful site and can represent Central-Eastern Europe, or rise problem about luck of observations. Please also be more specific about what you mean by "this" - you have not discussed this problem before.

P2 L54-55: Please add references to such studies.

Section 2: Please put information about the data gaps and gap-filling methods in this section for all sites. (Those of KAS is presented in capture of Figure 1, but please move it to this section). If there has been no gap in the data, please also notify.

P2 L61: Please add information about KAS data availability in this section.

P3 L94: Please specify which European network it belongs to.

P4 L122: Please be more specific about "periods of interest". You probably did not run footprints of the whole study period, but part of it?

P4 L126: Please justify your choice of "96-hours".

P5 L127, L137: You have used two different meteorological datasets (NCEP Reanalysis and ERA-Interim). Could you comment on those choices, how much differences there are, and whether the differences could possibly affect your analysis?

P5 L137: Please specify from which layers/level of air temperature data you have used (I could see that it is 2 m temperature from figures, but please also note in text). It could be good to specify it for soil humidity also in the beginning.

P5 L151-L152: Please explain what are included in the "biospheric fluxes" (biomass burning can also be considered as "biosphere" in some context).

P5 L152: The authors mention that ocean fluxes were negligible, but some (or actually quite many) MHD observations capture signals from ocean. Did you apply any specific filtering to such observations?

P6 L169: Please consider adding the following sentence before "It is apparent from Fig. 1b...": *Therefore, seasonal cycle amplitude and annual level is much lower for KAS than HUN.*

P7 L183-L185: How well do we know about seasonality of the anthropogenic emissions, and their contribution to the measured seasonal cycles at those sites? Please add references.

P8 L195: "uncertainty" → "standard error" (or deviation). In all the regression analysis, it was not very clear how you decide significances. For example, you mention that MHD peak-to-peak amplitude ($0.05 \pm 0.04$) is not significant, but KAS summer min. ($0.09 \pm 0.04$) is. If you have some additional quantitative values (e.g. p-value), that would help to better understand your arguments.

P9 L205-208: You already know the answer to this sentences. It would be better to go directly to the results, i.e. remove this sentence or rephrase.

Section 4.3: It seems that there are positive correlation between $CO_2$ amplitude and winter maximum, and negative correlation between $CO_2$ amplitude and summer minimum. Maybe you could consider starting discussion about those.

P13 L253: The authors meant to say that peak-to-peak amplitude is "low" in some years compared to average. Please rephrase the sentence to be more clear.

P14 L272-274: The authors explain that there is a clear differences in the direction of footprints, but I see strong influences from western Europe at both sites in Fig. 10. In addition, footprint of HUN show that the air is not coming much from northward directions. Could you explain these in a better way?

P20 L334-335: Please see my general comments about evaluation.

P20 L343-344: Please also rephrase "luck of strong biospheric signals". The station captures biospheric signals, but mostly from western direction. Maybe you could say "luck of biospheric signals from eastwards"?

Conclusion: It would be good to add conclusion about our actions to be taken. Shall we be more urgent in reducing anthropogenic emissions, if plants are suffering more from sever summers, and fertilization effect may be weakening?

Figure 4 and 5: Please put each of them into one figure (i.e. without panels), and use same colour codes.

Figure 7: Why do you compare only with KAS?

Figure 8, 9, 11: Is it necessary to show the all years? You could consider showing only those important years.

**Technical corrections**

P2 L40: "except of" → "except for", "Finish" → "Finnish"

P2 L56-59: Please use present tenses.

P5 L 137: "In order to calculate annual air temperature and..." → "In order to calculate annual regional air temperature and..." (It is good to specify in the beginning that you use these data for regional analysis, and not for site-level.)

P5 L150: "Wageningen University" → "Wageningen University and Research"

P20 L343: Please rephrase "...Hegyhatsal station because of larger footprint of Kasprowy Wierch and lack of" as "...Hegyhatsal station because of smaller footprint and...", i.e. be consistent with the subject.

---

## Referee Comment (RC2) · Anonymous Referee #2 · 9 Dec 2019

In their paper 'Signs of reduced biospheric activity with progressing global warming: evidence from long-term records of atmospheric CO2 mixing ratios in Central-Eastern Europe' Chmura et al. analyse time series of atmospheric CO2 from two stations in central Eastern Europe with respect to changes in the fluxes to the atmosphere causing changes in the annual amplitudes of the seasonal cycle of the atmospheric CO2 concentrations. They postulate that the reduction in the amplitude as seen by the observations at both the Kasprowy Wierch and the Hegyhatsal stations are caused mainly caused by a reduction in biospheric activity during periods of extreme weather,

ie. droughts.

The manuscript is mostly well written and concise, unfortunately too concise, which is the main problem of this manuscript.

The authors did not analyse all the potential contributions that can lead to changes in the seasonal cycle amplitude, they only discuss reduced photosynthesis for an increase in the summer minimum and reduced fossil emissions for a decrease in the winter maximum. They do not even mention all the other influencing factors such as changes in ecosystem respiration, changes in the growing season length, changes in land use, CO2 fertilisation (surely plays a role over the 24 year period) other changes in fossil fuel emission than reduced emissions from heating due to warmer winters.

The author also do not sufficiently support their hypothesis with other data sources. They only use results from CarbonTracker, however, integrated over the Transcom Europe region, which covers the whole geographical Europe extending eastwards to the Ural. This is a much larger region than what the footprints of the two analysed stations cover. Both the biospheric fluxes as well as the fossil fuel emissions used in Carbon-Tracker are available per grid-cell and analysing those data for the footprint regions of the stations would be more meaningful. Additional data sources (e.g. fossil fuel emissions from EDGAR, biospheric fluxes from the Global Carbon Project) for analysing the changes in emissions in Central-Eastern Europe are available, for instance, at the ICOS Carbon Portal.

Another aspect that is not discussed at all in the manuscript is how their findings of a reduced seasonal cycle amplitude relates to previous publications reporting an increased seasonal cycle amplitude in the northern hemisphere, e.g. Graven et al., 2013, and Forkel et al., 2015. Since this manuscript is rather contradicting these previous results, the changes in the seasonal cycle amplitude at the two stations subject of this paper need to be set in context with the other studies.

Detailed comments:

L34-35: It is not obvious why future climate predictions from numerical climate models rely on high-quality observations of atmospheric CO2 concentrations?

L 50ff: Why 'this' lack of representation, there hasn't been any mention of any lacks before.

L 121-122: Do you mean by 'periods of interest' that you have calculated footprints over the whole 24 year period? I assume that this is the period of interest.

L 133: What do you mean by constant 5 cell x 5 cell weighting field?

L134-135: Please explain the approach and not only provide a reference, especially if it is only similar and not the same approach!

L137: How about consistency in datasets when using ERA-interim for climate extreme detection and NCEP for footprint analysis?

L 144: Why do you use only the uppermost soil layer? Is this the soil moisture layer which affects plant water stress? Plants usually have much deeper roots and access to soil water at deeper layers.

References Graven et al., 2013. Enhanced Seasonal Exchange of CO2 by Northern Ecosystems Since 1960, Science 341 (6150), 1085-1089, DOI: 10.1126/science.1239207 Forkel et al., 2016. Enhanced seasonal CO2 exchange caused by amplified plant productivity in northern ecosystems, Science 351 (6274), 696-699, DOI: 10.1126/science.aac497

---

## Referee Comment (RC3) · Anonymous Referee #3 · 10 Dec 2019

In their study, Chmura et al. analyse long term CO2 mixing ratio records in two central European stations (KAS and HUN) to evaluate trends in the seasonality of CO2 and of the net surface fluxes. They find a decreasing trend in the seasonal cycle amplitude of CO2 and propose that this can be explained by decreasing fossil fuel (FF) emissions in winter. I find that the manuscript in the present form suffers from several problems, including poor referencing and discussion and some methodological issues. I explain my points below.

I find the introduction unfortunately too poor in terms of framing the current study in

the previous works evaluating trends in the seasonal cycle amplitude of CO2 in the northern Hemisphere (Graven et al. 2013, Forkel et al, 2015, Piao et al. 2017, Yin et al. 2018, to name only a few). The introduction focuses instead on crop studies (Schauberger) and the impact of extreme events. While the extreme summers are relevant for this study, I do not understand the choice to discuss crop productivity or the underestimation of summer impacts by land surface models (not used in this study). I recommend a full restructuring and appropriate referencing of the introduction to address clearly: - why the authors analyse the amplitude in the seasonal cycle of CO2 mixing ratios (SCA) and what is the current debate on trends and drivers of SCA - why is central Europe a region of interest and why do they claim that central Europe is poorly represented (compared to other regions in the globe, it has much higher density of stations...) - what is the value of long term monitoring sites used in this study - what novel aspects are brought by this study - perhaps one sentence or two on key findings

Likewise, the conclusions fail to set the current findings in contrast with the previous studies, especially since they are to some extent contradictory with some studies (e.g. Graven, Forkel). Still, Penuelas et al., 2017 has pointed that a slow down in SCA at Barrow was observed, and this could be possibly due to increasingly negative impacts of extreme summers. Yin et al., 2018 also found a strengthening of the negative relationship of SCA with temperature.

There are in addition some points that I find problematic:

1) Lines 231-234 Why do the authors immediately conclude that the reduction in the winter peak is due to reduction in fossil fuel emissions only? Is it not possible that biospheric processes play a role? The authors could answer this question by using FF emission data and transporting the fluxes forward to evaluate the contribution of FF to the seasonal cycle amplitude of CO2 on this site. I would argue that the analysis does not settle the attribution to either anthropogenic or biospheric fluxes.

2) The comparison with CTE could in part address this issue, but unfortunately I find

several problems with the methodology. The authors compare the results of site-level SCA with continental averaged CTE surface fluxes, which I do not think it correct. First, the authors do not define how the European continent is defined. Secondly, the authors should only compare the fluxes from CTE that are within the site's footprint, which the authors then show in Figure 10 to be quite variable, and to not cover the full European continent. I think the appropriate method to attribute changes in SCA to FF or Biospheric fluxes would be to transport forward the fluxes from CTE in order to calculate the resulting concentrations at HUN and KAS). This methodology has been used for example by Piao et al., 2017 to perform attribution in SCA changes from factorial simulations by land surface models. Thirdly, the authors use only one dataset for FF emissions and one atmospheric inversion system. However, Gaubert et al. 2019 has shown that there is large disagreement in hemispheric fluxes between different inversions systems (and smaller regions should be even more difficult to constrain), and that a large fraction of the disagreement between inversions could be attributed to the FF emission data sets used. Therefore, it would be advisable to include more atmospheric inversion datasets to obtain an uncertainty range for surface fluxes. Finally, in Fig. 7 the authors compare apples and oranges: in-situ $CO_2$ mixing ratios in ppm/yr with continental net biospheric exchange. By doing this, the authors assume that trends in [$CO_2$] SCA are directly linked to net annual $CO_2$ exchange, but trends in SCA could be found even if the net annual balance would not change, for example if increased uptake in summer would be offset by increased release in autumn and winter (see Piao et al. 2008 and Figure S9 in Bastos et al. 2019 ACP). Decreasing winter amplitude could also be explained by increased photosynthesis under warmer winters (which the authors indicate in Fig. 11). As mentioned above, the only way(s) to make the attribution to different processes would be to translate $CO_2$ surface fluxes into concentration space using an atmospheric transport model, or else to invert $CO_2$ concentrations into fluxes, and comparing the site footprints with CTE.

3) The use of statistics. The authors overstate confidence in some results that are non-significant, e.g. Lines 240-243 "as well as the growing net $CO_2$ flux of the continental

biosphere" - which is 0.03 +- 0.03, and therefore non-significanly different than zero. On the other hand, when discussing trends in Mace Head the authors state that trends are not discernible (Lines 198-199), but the value is 0.05+-0.04, which could be considered significantly increasing.

4) The discussion of extremes is generally interesting, but I find a similar problem as with the comparison with CTE above. First, the authors present results for the climate anomalies in the whole European region. I do not think Figures 8 and 9 should be in the main text, but they can be provided in supplement. I would have found it more interesting to see the anomalies in T and water content from the sites' footprints for 2003 and 2010 to understand how representative they are, and whether one can see differences between stations because of somewhat different footprints.

The manuscript is well structured but I find the writing sloppy, with many grammar errors/inconsistencies which sometimes make it difficult to understand the message. Examples of sloppy/unclear writing include "an aggregated gridded spatial maps of area of influences", "annual air temperature and soil anomalies for summer and winter seasons", "periods of interests", "as functions of time". Moreover there are many examples where "the" is missing, and other small grammar inconsistencies are found.

Line 242: shouldn't the rate of FF emissions be -0.7 rather than 0.7?

---

## Editor Comment (EC1) · Janne Rinne (Editor) · 30 Jan 2020

Dear Łukasz Chmura et al., While reading the discussion and rereading the paper, I started wondering the source of the decrease in the seasonal amplitude of CO2 mixing ratios at the two East European sites. In the paper it is stated that this decrease is equally due to lowering of seasonal maxima and increase in minima (Figures 4 and 5). However, these minima and maxima are derived as differences from long term mean (i.e. average over the seasonal cycle), and thus being affected by these minima and maxima. Thus, it seems to me that this could be a case of circular reasoning. Could

you comment on this issue? Best, Janne Rinne

---

## Author Comment (AC1) · 12 Feb 2020

Question from the Editor:

"While reading the discussion and rereading the paper, I started wondering the source of the decrease in the seasonal amplitude of CO2 mixing ratios at the two East European sites. In the paper it is stated that this decrease is equally due to lowering of seasonal maxima and increase in minima (Figures 4 and 5). However, these minima and maxima are derived as differences from long term mean (i.e. average over the

**Discussion** paper

seasonal cycle), and thus being affected by these minima and maxima. Thus, it seems to me that this could be a case of circular reasoning".

Reply:

We are not sure whether we understood your comment correctly. Let me reiterate our approach to the data originating from the sites discussed in the manuscript: (i) the available daily CO2 mixing ratio data were subject to smoothing using CCGvu 4.40 routine (Thoning et al., 1989) resulting in the smoothed records (cf. Fig. 1) (ii) CCGvu 4.40 routine calculated also trend line for each record (also shown in Fig.1b) (iii) detrended records shown in Fig. 2 were calculated by subtracting, point by point (i.e. on daily basis), the trend line from the smoothed record available for each monitoring site. Therefore, no averaging over seasonal cycle, as suggested by your comment, was applied in the detrending procedure. Thus, we are confident that no circular argument is involved in the reasoning pursued in our manuscript.

---

## Author Comment (AC2) · 19 Feb 2020

Subject 1: "Authors conclude that the changes in the SCA have been driven only by an increase in biospheric fluxes in summer and fossil fuel emissions in winter."

Response:

In Xu et al., 2015 (https://doi.org/10.3389/fpls.2015.00701) one can read a summary of the respiration and photosynthesis reaction to elevated CO2 level and it should be taken into the account when analyzing the CO2 record. According to Tomczyk et al.

[Figure]

(2018) (doi:10.1007/s00704-018-2450-4), the season is getting longer by 0.5 days/year what makes it longer by 10days on average during the last 20 years. It is consistent with MOPIT satellite estimates or other hyperspectral methods. According to EEA Report No 10/2017, the land-use change is on the constant level and hasn't substantially changed during the last 20 years contributing to total anthropogenic $CO_2$ release rate by approx. 15% (IPCC, 2019). There is no statistical evidence of the acceleration of land-use change in the recent 20 years, which would also contribute to SCA. Zhu et al. (2016) (https://doi.org/10.1038/nclimate3004) is suggesting that there are significant changes in LAI (leaf area index) over Europe. According to Haverd et al. (2020) (https://doi.org/10.1111/gcb.14950), we expect that during the last 20 years rate of photosynthetic activity might rise by 5%. It should be included in our discussion and it will be.

Certainly, we are aware of the changes occurring on the market of fossil fuel consumption and its influence on carbon dioxide budget (Agora Energiewende and Sandbag, 2018). We are going to discuss it deeper, especially the changes in its seasonal pattern. Most of the databases do not include updated, high temporal resolution emission rates, however, we will try to look at the relation of temperature-energy consumption to retrieve the changes and we will add 14C analysis which helps in better understanding of fossil-fuel emission trend.

References:

Agora Energiewende and Sandbag (2018): The European Power Sector in 2017. State of Affairs and Review of Current Developments.

EEA Report No 10/2017, Landscapes in transition. An account of 25 years of land cover change in Europe, ISSN 1977‑8449, European Environment Agency, 2017, doi:10.2800/81075

Haverd, V, Smith, B, Canadell, JG, et al. Higher than expected $CO_2$ fertilization inferred from leaf to global observations. Glob Change Biol. 2020; 00: 1– 13.

https://doi.org/10.1111/gcb.14950

IPCC, 2019: Climate Change and Land: an IPCC special report on climate change, desertification, land degradation, sustainable land management, food security, and greenhouse gas fluxes in terrestrial ecosystems [P.R. Shukla, J. Skea, E. Calvo Buendia, V. Masson-Delmotte, H.-O. Pörtner, D. C. Roberts, P. Zhai, R. Slade, S. Connors, R. van Diemen, M. Ferrat, E. Haughey, S. Luz, S. Neogi, M. Pathak, J. Petzold, J. Portugal Pereira, P. Vyas, E. Huntley, K. Kissick, M. Belkacemi, J. Malley, (eds.)]. In press.

Tomczyk, A. M., Szyga-Pluta, K. "Variability of Thermal and Precipitation Conditions In the Growing Season In Poland In the Years 1966–2015." Theoretical and applied climatology, v. 135,.3-4 pp. 1517-1530. doi:10.1007/s00704-018-2450-4

Xu Z, Jiang Y and Zhou G (2015), Response and adaptation of photosynthesis, respiration, and antioxidant systems to elevated CO2 with environmental stress in plants., Front. Plant Sci. 6:701. doi:10.3389/fpls.2015.00701

Zhu, Z., Piao, S., Myneni, R. et al. Greening of the Earth and its drivers. Nature Clim Change 6, 791–795 (2016). https://doi.org/10.1038/nclimate3004

Subject 2: "If the increase in biospheric fluxes is a major cause in the increase of summer minimum atmospheric CO2, it is due to the reduction of photosynthesis or increase in respiration?"

Response:

We thank the reviewer for raising this important question.

Using the atmospheric observations of CO2 alone cannot yield an answer to that question. Support either from other tracers or isotopes (see e.g. Zimnoch et al. 2012) might yield additional information, but the results are characterized by high uncertainty. Another option is to use modeling frameworks, but current state-of-the-art models do not even provide sufficiently accurate estimates of the fluxes. For example, an overview of the global vegetation models available in Global Carbon Budget 2018 shows that in

general, the global models performance is medium for GPP and ecosystem respiration, and low-to-medium for estimating NEE (Le Quere et al., 2018).

In our paper, we wanted to keep the analysis primarily data-driven, therefore we only discuss temporal patterns of net biospheric flux (or net ecosystem exchange, NEE). This approach will be underlined in the revised text.

References:

Le Quéré, C., Andrew, R. M., Friedlingstein, P., Sitch, S., Hauck, J., Pongratz, J., Pickers, P. A., Korsbakken, J. I., Peters, G. P., Canadell, J. G., Arneth, A., Arora, V. K., Barbero, L., Bastos, A., Bopp, L., Chevallier, F., Chini, L. P., Ciais, P., Doney, S. C., Gkritzalis, T., Goll, D. S., Harris, I., Haverd, V., Hoffman, F. M., Hoppema, M., Houghton, R. A., Hurtt, G., Ilyina, T., Jain, A. K., Johannessen, T., Jones, C. D., Kato, E., Keeling, R. F., Goldewijk, K. K., Landschützer, P., Lefèvre, N., Lienert, S., Liu, Z., Lombardozzi, D., Metzl, N., Munro, D. R., Nabel, J. E. M. S., Nakaoka, S., Neill, C., Olsen, A., Ono, T., Patra, P., Peregon, A., Peters, W., Peylin, P., Pfeil, B., Pierrot, D., Poulter, B., Rehder, G., Resplandy, L., Robertson, E., Rocher, M., Rödenbeck, C., Schuster, U., Schwinger, J., Séférian, R., Skjelvan, I., Steinhoff, T., Sutton, A., Tans, P. P., Tian, H., Tilbrook, B., Tubiello, F. N., van der Laan-Luijkx, I. T., van der Werf, G. R., Viovy, N., Walker, A. P., Wiltshire, A. J., Wright, R., Zaehle, S., and Zheng, B.: Global Carbon Budget 2018, Earth Syst. Sci. Data, 10, 2141–2194, https://doi.org/10.5194/essd-10-2141-2018, 2018

Zimnoch M., Jelen D., Galkowski M., Kuc T., Necki J., Chmura L., Gorczyca Z., Jasek A., Rozanski K., (2012): Partitioning of atmospheric carbon dioxide over Central Europe: insights from combined measurements of CO2 mixing ratios and their carbon isotope composition, Isotopes in Environmental and Health Studies, DOI:10.1080/10256016.2012.663368

Subject 3: "How about feedbacks, e.g. fertilization effects (only one sentence about fertilization in the conclusion)?"

[Figure]

Response:

Considerations about feedback processes, such as, for example, the effect of fertilizing the atmosphere with carbon dioxide, is not the purpose of the analysis presented here. Of course, the authors are aware of the existence of such processes as well as the complex nature of changing the mechanisms that control these processes in the context of progressive climate change. However, the beginning of the 21st century indicates that the fertilization process is weakening. Yin et al. (2018) point out that the correlation between CO2 seasonal cycle amplitude and the temperature became negative around the year 2000 at most northern stations. It seems to confirm a limit to the "warmer spring – bigger carbon sink" mechanism. This finding highlights a dynamic temperature sensitivity of the terrestrial ecosystem to climate warming and cautions the use of current carbon-climate response to constrain future projections. This issue will be elaborated in the revised version.

References:

Yin, Y., Ciais, P., Chevallier, F., Li, W., Bastos, A., Piao, S., et al. (2018). Changes in the response of the Northern Hemisphere carbon uptake to temperature over the last three decades. Geophysical Research Letters, 45, 4371–4380. https://doi.org/10.1029/2018GL077316

Subject 4: "The authors have concluded that the trends and events of the changes in the SCA are associated with climate conditions, but did not discuss in detail the effects of anthropogenic actions, such as land-use change and policy or economic conditions."

Response:

We thank the reviewer for pointing this out. We would like to point out that the effects of anthropogenic actions mentioned by the reviewer have been taken into the account in our analysis indirectly, by comparing the analyzed SCAs to multi-annual patterns in European fossil fuel emissions of CO2. These take into account land-use change as

well as economic conditions.

However, we agree that the discussion on patterns of anthropogenic emissions needs to be expanded in the revised manuscript.

Subject 5: "References to show that fossil fuel emissions have decreased is missing (CTE and thus its prior is from a model). Also, the seasonal cycle of anthropogenic emissions could have changed as well – warmer winter and more severe summer could cause a shift in fossil fuel emissions from winter heating to summer cooling."

Response: a) "References to show that fossil fuel emissions have decreased is missing (CTE and thus its prior is from a model)."

We would like to thank the reviewer for pointing the missing reference.

We would like to point out that the emissions used in CTE are based on the bottom-up statistical approach and not from a model; specifically, these were not optimized by the inversion system. From CarbonTracker system description (https://www.carbontracker.eu/documentation_cte2018.pdf): "The fossil fuel emission inventory used in CarbonTracker Europe is the one constructed for the CARBONES project by USTUTT/IER. It uses emissions from the EDGAR 4.2 database together with country and sector specific time profiles derived by IER. A detailed description of the construction of the product is found here. The global total emissions for 2000-2017 were scaled to the global totals used in the GlobalCarbon Budget 2017. An individual trend per continent/Transcom region was applied in this scaling." The text will be modified to clarify the source of fossil-fuel data.

b) "In addition, the seasonal cycle of anthropogenic emissions could have changed as well – warmer winter, and more sever summer could cause shift in fossil fuel emissions from winter heating to summer cooling."

Thank you for pointing it out. Indeed, there is evidence that the pattern of CO2 emissions is changing in some countries. However, this response is far from uniform. For

example, in Poland, the power production from combustible sources (most of the coal power plants) in winter has remained stable since 2000 (approx. 12.5 GWh/month), while the summer power production values have increased by approx. 15%, from 9.2 GWh/month in June 2000 to 10.6 in June 2019). On the other hand, in the largest EU economy – Germany – the amplitude of power generation from combustible sources has not changed significantly (Jan-Jun amplitude of 9.0 GWh in 2000 and 9.3 GWh for 2018), and at the same time, the overall power production from fossil fuels fluctuated without showing any clear trend (annual means between 29-34 GWh/month), which is probably related to larger fraction of power used by industry and not individual users. Source: "IEA Monthly electricity statistic, Revised Historical Data", version from November 2019. Overall, while the "winter-heating summer-cooling" effect indeed occurs in certain areas of Europe, this is definitely not the case throughout the continent. We will analyze this in more depth in our revised manuscript.

Subject 6: "The evaluation with flux sources is done by comparing flux estimates from a model. As footprints of summer and winter are shown to dominate from Central-Western Europe, how much about Eastern Europe can you say? The CTE results are from the whole of Europe, so how do you incorporate your arguments considering the trajectories? To be more precise, source contributions should be calculated by running trajectories or transport models for each source separately."

Response:

We thank the reviewer for the recommendation.

We would like to first point out that the source of flux estimates is not a model, but rather bottom-up statistical data supported by model results. See the remark earlier in the text for details (response to subject 5a).

In our approach, we have decided to limit the usage of the model data to a minimum and focus on atmospheric observations instead. In particular, we did not envisage the expansion of the trajectory analysis, as this would shift the emphasis from data- to

model-driven analysis. Issues like transport error, prior source attributions and lateral boundary conditions need to be carefully analyzed and addressed. Expanding the analysis as per reviewer suggestions is a significant project that deserves a paper of its own, and it is beyond the scope of this study.

Instead, by our selection of measurement sites and data selection procedures (night-time data only, filtered for local influences by procedure described in Necki et al., (2003) and Chmura, (2010), we are interpreting the regional background representative of most of Central-Eastern Europe, even if this is not apparent from our footprint analysis. In the current version, the footprints represent only the frequency of single-member trajectories passing through any given grid cell. This approach is not sufficient to properly account for the areas of source influences, as we only simulate a single trajectory and do not take PBL height into the account. To do this, statistical ensemble models would be the appropriate tools (e.g. Stochastic Time-Inverted Lagrangian Transport Model STILT, Lin et al. (2003)). This will be clarified in the revised manuscript.

References:

Chmura L., Gazy cieplarniane w atmosferze Polski Południowej: zmienność czasowo-przestrzenna w okresie 1994-2007 (in Polish), PhD thesis, AGH – University of Science and Technology, 2010. Lin, J. C., Gerbig, C., Wofsy, S. C., Andrews, A. E., Daube, B. C., Davis, K. J., and Grainger, C. A. (2003), A near‐field tool for simulating the upstream influence of atmospheric observations: The Stochastic Time‐Inverted Lagrangian Transport (STILT) model, J. Geophys. Res., 108, 4493, doi:10.1029/2002JD003161, D16. Necki, J.M., Schmidt, M., Rozanski, K., Zimnoch, M., Korus, A., Lasa, J., Graul, R. and Levin, I. Six year Record of Atmospheric Carbon Dioxide and Methane at a High-altitude Mountain Site in Poland. Tellus Ser. B 55: 94–104, 2003.

Subject 7: "Using results from only one model in evaluation should be done with caution. Any models have their own features, and results may be biased. In addition, if

those studied sites were assimilated in the CTE inversion (i.e. has influence on the flux results), the evaluation is not independent of the measurements. Please specify and justify the choice, and if possible, use other sources, such as results from Global Carbon Project for biospheric fluxes and UNFCCC reports for anthropogenic emissions as additional source of evaluation."

Response:

We thank the reviewer for making this point.

We wholeheartedly agree that the results of the models need to be treated with care, and we will revise the text in order to underline the potential biases and known issues of the CT-Europe system.

We have selected CarbonTracker-Europe (CTE), as it is (i) a state-of-the-art modelling system, (ii) it is representative of the mean of other similar systems in our latitudes (see Table A3 and Figure B3 in Global Carbon Budget 2018, Le Quere et. al. (2018), and (iii) data from CTE are readily available. We will also underline the reasons for selecting CTE in the revised manuscript.

Both KAS and HUN stations were indeed assimilated by the CTE system. However, we do not believe this is an issue for our analyses for two reasons: first, we are focusing on the regional phenomena, specifically by analyzing the net $CO_2$ fluxes from the complete TRANSCOM-Europe area. Second, we use the model only to test if the net biospheric prediction over the TRANSCOM area corroborates our results. We do not have the means of directly validating the model against observations, as this would require a forward run of the underlying transport model using the optimized fluxes. We will clarify this relation in the revised manuscript.

References:

Le Quéré, C., Andrew, R. M., Friedlingstein, P., Sitch, S., Hauck, J., Pongratz, J., Pickers, P. A., Korsbakken, J. I., Peters, G. P., Canadell, J. G., Arneth, A., Arora, V. K.,

Barbero, L., Bastos, A., Bopp, L., Chevallier, F., Chini, L. P., Ciais, P., Doney, S. C., Gkritzalis, T., Goll, D. S., Harris, I., Haverd, V., Hoffman, F. M., Hoppema, M., Houghton, R. A., Hurtt, G., Ilyina, T., Jain, A. K., Johannessen, T., Jones, C. D., Kato, E., Keeling, R. F., Goldewijk, K. K., Landschützer, P., Lefèvre, N., Lienert, S., Liu, Z., Lombardozzi, D., Metzl, N., Munro, D. R., Nabel, J. E. M. S., Nakaoka, S., Neill, C., Olsen, A., Ono, T., Patra, P., Peregon, A., Peters, W., Peylin, P., Pfeil, B., Pierrot, D., Poulter, B., Rehder, G., Resplandy, L., Robertson, E., Rocher, M., Rödenbeck, C., Schuster, U., Schwinger, J., Séférian, R., Skjelvan, I., Steinhoff, T., Sutton, A., Tans, P. P., Tian, H., Tilbrook, B., Tubiello, F. N., van der Laan-Luijkx, I. T., van der Werf, G. R., Viovy, N., Walker, A. P., Wiltshire, A. J., Wright, R., Zaehle, S., and Zheng, B.: Global Carbon Budget 2018, Earth Syst. Sci. Data, 10, 2141–2194, https://doi.org/10.5194/essd-10-2141-2018, 2018

Subject 8: "P2 L42-43: The advantage of GHG observations are useful not only for studying last decades, but also present and future. It is true that "currently available" observations can only see the past, but findings from those studies can be used for future predictions as well."

Response:

That is, of course, true, so we added one more sentence and now the relevant part of the text may look like that:

The availability of dense GHG observation networks allows scientists to analyze both long-term trends and anomalies in carbon balance as well as the response of the biosphere to changes in climate observed during the last decades. The advantage of GHG observations is useful not only for studying the last decades but also present and future. Currently available observations can only see the past, but findings from those studies can be used for future predictions as well.

Subject 9: "P2 L48-49: What was their main conclusion?"

[Figure]

Response:

The first one is that global climate models underestimate climate extremes (2003 European heatwave event can be useful as an example). The second one pointed out that the 2003 heatwave event significantly reduced the primary productivity of the European biosphere. The text will be modified accordingly.

Subject 10: "P2 L50-53: Please rephrase. It is not clear whether you wish to say that KAS is a useful site and can represent Central-Eastern Europe, or rise problem about the lack of observations. Please also be more specific about what you mean by "this" - you have not discussed this problem before."

Response:

The whole section will be modified as follows:

The lack of proper representation of Central-Eastern Europe in present GHG observation networks is partly compensated by the Kasprowy Wierch greenhouse gas monitoring station. The station is located in the High Tatras mountain range of southern Poland, at the level of 1989 m a.s.l. This high elevation of the measuring point and lack of strong $CO_2$ sources in the direct vicinity of the station assure that the measured $CO_2$ signal is representative for Central-Eastern European background.

Subject 11: "P2 L54-55: Please add references to such studies ('Long-term time series of $CO_2$ and $CH_4$ available for this site, in combination with relevant data from other stations, especially the Mace Head baseline station (monitoring greenhouse gas characteristics of maritime air masses entering Europe), enabled to study the impact of continental sources and sinks of GHGs on the observed atmospheric levels of those gases in the interior of the European continent.')."

Response:

A proper reference (Rozanski et al., 2016) will be added to the text.

[Figure]

Rozanski K., Chmura L., Galkowski M., Necki J., Zimnoch M., Bartyzel J., O'Doherty S., Monitoring of Greenhouse Gases in the Atmosphere: a Polish Perspective PAPERS on GLOBAL CHANGE, 23, 111–126, 2016, DOI: 10.1515/igbp-2016-0009

Subject 12: "Section 2: Please put information about the data gaps and gap-filling methods in this section for all sites. (Those of KAS is presented in the capture of Figure 1, but please move it to this section). If there has been no gap in the data, please also notify."

Response:

Appropriate information will be added to the text in section 2.1, 2.2 and 2.3 for Kasprowy Wierch, Mace Head and Hegyhatsal respectively:

"In the entire period of operation of the greenhouse gas monitoring station at Kasprowy Wierch (from September 1994 to December 2018), the data coverage amounts is equal to 71% of the total working time. At KASLAB station due to instrument malfunction, there was one longer period with lack of measurement. Data between July and December 1997 have been gap-filled with a mean annual cycle calculated from period 1996 – 2018, added to a long-term linear trend."

"In the years 1994-2017, CO2 concentration data for Mace Head stations cover 81% of the period discussed here. There were no major technical problems at the station during this time, so there was no need to fill gaps in the measurement data."

"Available CO2 concentration data cover 85% of the given period. In the case of Hegyhatsal station also there were no major technical problems at the station during this time, so there was no need to use any procedures to fill gaps in the measurement data."

Subject 13: "P2 L61: Please add information about KAS data availability in this section."

Response:

[Figure]

The following information will be added:

"The data for both stations were retrieved from the GLOBALVIEW-CO2 NOAA ESRL Carbon Cycle Cooperative Global Air Sampling Network (http://www.esrl.noaa.gov/gmd/dv/data/ – Dlugokencky et al., 2018)."

Subject 14: "P3 L94: Please specify which European network it belongs to."

Response:

The tower is the NOAA/CMDL global air sampling network site (site code: HUN) (Conway et al., 1994). It was part of the CHIOTTO network. This network was nowadays replaced by ICOS, and HUN site does not belong to it. The text will be modified accordingly

References

Conway, T. J., Tans, P. P., Waterman, L. S., Thoning, K. W., Kitzis, D. R., Masarie, K. A., and Zhang, N. (1994), Evidence for interannual variability of the carbon cycle from the National Oceanic and Atmospheric Administration/Climate Monitoring and Diagnostics Laboratory Global Air Sampling Network, J. Geophys. Res., 99 (D11), 22831– 22855, doi:10.1029/94JD01951.

Subject 15: "P4 L122: Please be more specific about "periods of interest". You probably did not run footprints of the whole study period, but part of it?"

Response:

The sentence will be rephrased in the revised manuscript:

"Assessment of the area of influence (footprints) for Kasprowy Wierch and Hegyhatsal stations has been carried out for the three month periods (June, July and August) 2003 and 2010 with hourly resolution as described in the figure captions. Each frequency plot was created based on 2208 individual trajectories."

Subject 16: "P4 L126: Please justify your choice of "96-hours"."

Response:

The trajectory length selected for analysis depends on the specific application. In general, the shorter the time, the better the accuracy. However, the too-short trajectory will represent a smaller area surrounding the reception point. While 24-48 hours is a good choice for the identification of possible pollution sources located close to the reception point, a statistical regional analysis like frequency of clustering requires a longer time. The length of 96 hours was assumed based on the recommendation in Hysplit FAQ (https://www.arl.noaa.gov/hysplit/hysplit-frequently-asked-questions-faqs/backward-trajectories-starting-at-a-rural-east-coast-site/), (Rolph et al., 1990).

References:

Rolph, G. D., Draxler R. R., "Sensitivity of Three-Dimensional Trajectories to the Spatial and Temporal Densities of the Wind Field." Journal of Applied Meteorology (1988-2005) 29, no. 10 (1990): 1043-054. Accessed February 14, 2020. www.jstor.org/stable/26185789.

Subject 17: "P5 L127, L137: You have used two different meteorological datasets (NCEP Reanalysis and ERA- Interim). Could you comment on those choices, how much differences there are, and whether the differences could possibly affect your analysis?."

Response:

NCEP data was chosen to drive the Hysplit model for several reasons: it was readily available for both periods of interest (2003, 2010), had low disk space requirements and was already well established. While it is true that ERA-Interim (or even newer datasets like ERA5) represent the actual atmospheric state conditions better, we believe that the difference between calculated footprint areas would not be significant enough to

warrant extra work necessary for the conversion of data.

At the same time, we recognize that the overall quality of the ERA-Interim dataset to be higher, which is why we used it in the - more critical to the discussion - analysis of the state of the soil during the analyzed period.

Subject 18: "P5 L137: Please specify from which layers/level of air temperature data you have used (I could see that it is 2 m temperature from figures, but please also note in the text). It could be good to specify it for soil humidity also in the beginning."

Response:

Thank you for pointing this out. Indeed, the 2m air temperature was used for the analysis. We modified the text to include the information as per your suggestion.

Following the comment by another reviewer, the calculation of soil humidity was also changed from using the first level only to the weighted arithmetic mean of soil humidity profile obtained by using four levels available in ERA-Interim data. The weighting function was scaled using the depth of soil layers. A detailed description was added in section 3.3.

Subject 19: "P5 L151-L152: Please explain what is included in the "biospheric fluxes" (biomass burning can also be considered as "biosphere" in some context)."

Response:

In the CarbonTracker model, the biospheric module is based on the Simple-Biosphere-Model-Carnegie-Ames Stanford Approach (SiBCASA) model, and optimizes the net ecosystem exchange (NEE), "derived directly from Gross Primary Production (GPP) and ecosystem respiration (R) from SiBCASA" (https://www.carbontracker.eu/documentation_cte2018.pdf).

Biomass burning emissions are included in the separate fire module in the Carbon-Tracker system, driven by GFEDv4 (Global Fire Emission Database v4) and output

from SiBCASA biosphere model (i.e. seasonally changing vegetation and soil biomass stocks). The fire emissions calculated in CarbonTracker over the Transcom Region Europe were never higher than 0.02 PgC /year (usually 0.01 Pg/C) and did not show significant variability. We have therefore excluded the emissions from biomass burning from our analysis.

Subject 20: "P5 L152: The authors mention that ocean fluxes were negligible, but some (or actually quite many) MHD observations capture signals from the ocean. Did you apply any specific filtering to such observations?"

Response:

The referee is of course right – MHD station in the major part captures the signal from the Atlantic Ocean (clean section for this site is in the range for 180o to 300o). It is a costal station so the signal from the ocean-atmosphere flux is evident. But the aim of our article is to show how the atmosphere reacts to changes over time in CO2 exchange fluxes between land and the atmosphere itself. In that context, the concentration of carbon dioxide measured at MHD station is a kind of input data for our analysis. That is the reason why we decided to choose the European TrnasCom region for biospheric and fossil-fuel CO2 fluxes. In that range ocean-atmosphere flux is negligible.

Subject 21: "P6 L169: Please consider adding the following sentence before "It is apparent from Fig. 1b...": Therefore, seasonal cycle amplitude and annual level is much lower for KAS than HUN."

Response:

Good point. Thank you. The sentence will be added to the text.

Subject 22: "P7 L183-L185: How well do we know about seasonality of the anthropogenic emissions and their contribution to the measured seasonal cycles at those sites?"

Response:

The seasonality of the anthropogenic emissions is discussed partially in the answer to the previous comment of the reviewer (see in particular response to subject 5b) and will be further discussed in the revised manuscript. A limited-time study by Zimnoch et al. (2012) has used isotopic information to derive monthly means of fossil and biogenic additions to the large-scale background (for 2008-2009) and concluded that the anthropogenic addition to the signal at KAS varied between 0 to 4 ppm, with maximum values observed in winter. The revised text will contain this additional information.

References:

Zimnoch M., Jelen D., Galkowski M., Kuc T., Necki J., Chmura L., Gorczyca Z., Jasek A., RóÅijański K., Partitioning of atmospheric carbon dioxide over Central Europe: insights from combined measurements of CO2 mixing ratios and their carbon isotope composition, Isotopes in Environmental and Health Studies, Vol. 48, Issue: 3, 421-33. doi: 10.1080/10256016.2012.663368., 2012

Subject 23: "P8 L195: "uncertainty" → "standard error" (or deviation). In all the regression analysis, it was not very clear how you decide significances. For example, you mention that MHD peak-to-peak amplitude (0.05±0.04) is not significant, but KAS summer min. (0.09±0.04) is. If you have some additional quantitative values (e.g. p-value), that would help to better understand your arguments."

Response:

Dear reviewer, according to the "International vocabulary of metrology – Basic and general concepts and associated terms (VIM), 2012" ant other GUM standards, uncertainty seems to be the more correct expression in this case.

Thank you very much for drawing attention to the study of statistical hypotheses. The p-values will be quoted in the text where necessary. However, in the case of numerous samples, such as this, analysis comes down to a comparison of values (for example,

curve slope coefficients) and their uncertainty. If the value differs from zero by less than two sigma, it is obvious that the p-value will be significantly greater than 0.05 and the hypothesis of a difference of the coefficient from zero can be rejected. Therefore, in our approach, we did not consider giving p-values. We also plan to graphically mark the uncertainty of the fits on the charts.

Subject 24: "P9 L205-208: You already know the answer to these sentences. It would be better to go directly to the results, i.e. remove this sentence or rephrase."

Response:

We thank the reviewer for pointing this out. We have revised the discussed sentence, and now it reads: "We believe that these decreasing amplitudes are a result of two main physical effects: (i) higher mixing ratios of $CO_2$ recorded during peaks of summer seasons, and/or (ii) lower $CO_2$ mixing ratios recorded during winter periods."

Subject 25: "P13 L253: The authors meant to say that peak-to-peak amplitude is "low" in some years compared to average. Please rephrase the sentence to be more clear."

Response:

The first paragraph of this section has been changed as suggested and now it has the following form: The average $CO_2$ seasonal cycle amplitude at Kasprowy Wierch and Hegyhatsal station is equal to 16.6 ± 0.5 ppm and 32.8 ± 1.0 ppm respectively. The years in which $CO_2$ seasonal cycle amplitude was lower than 14 ppm for Kasprowy Wierch and 30 ppm for Hegyhatsal were considered to be anomalously low values of this parameter. As discussed above, low peak-to-peak amplitudes of the seasonal $CO_2$ cycle recorded in a given year at stations located in Central-Eastern Europe indicate weak biospheric $CO_2$ sink during summer and reduced fossil-fuel flux of $CO_2$ into the regional atmosphere during winter, both considered within the area of influence (footprint) of the given station. Weak biospheric sink during summer may result from heat waves and/or reduced availability of moisture in the soil profile. As can be seen

in Fig. 4a, there were five years in the data record available for Kasprowy Wierch, with peak-to-peak amplitudes of the seasonal CO2 cycle smaller than 14 ppm: 2003, 2010, 2012, 2014, 2015. The years 2003, 2012 and 2015 also stand out as years with the lowest peak-to-peak amplitude in the CO2 record of the Hegyhatsal station. Also, carbon dioxide seasonal cycle amplitude below 30 ppm was also recorded in 2002 and 2008 at this station (Fig. 3).

Subject 26: "P14 L272-274: The authors explain that there are clear differences in the direction of footprints, but I see strong influences from western Europe at both sites in Fig. 10. In addition, footprint of HUN show that the air is not coming much from northward directions. Could you explain these in a better way?"

Response:

Analysis of the overall shape of footprint calculated for KAS and HUN stations may suggest that both stations were influenced by emissions originating from areas impacted by heatwave located in western Europe. However, most of the HUN footprint representing France, northern Italy and the United Kingdom have frequency values below 1% constituting a small contribution to the station signal. The shape of the area representing more than 1% of trajectories frequency, in the case of HUN station, represents a much smaller region impacted by a heatwave. Appropriate changes will be made in the text.

Subject 27: "P20 L343-344: Please also rephrase "luck of strong biospheric signals". The station captures biospheric signals, but mostly from the western direction. Maybe you could say "luck of biospheric signals from eastwards"?"

Response:

We thank the reviewer for the suggestion. We have modified the sentence in question. It now reads: "From this perspective, high-altitude mountain station Kasprowy Wierch appears to be better suited for tracing subtle, long-term changes of carbon cycle over

the European continent than low-altitude Hegyhatsal station, for which a smaller footprint and stronger local signals influencing CO2 mixing ratios in the PBL effectively mask larger-scale effects."

Subject 28: "Conclusion: It would be good to add a conclusion about our actions to be taken. Shall we be more urgent in reducing anthropogenic emissions, if plants are suffering more from sever summers, and fertilization effect may be weakening?"

Response:

We thank the reviewer for this suggestion. We believe, however, that the urgency of the matter is obvious for readers of the ACP Journal and further emphasis is not needed.

Subject 29: "Figures 4 and 5: Please put each of them into one figure (i.e. without panels), and use the same color codes."

Response:

Figures 4 and 5 will be changed, to follow your suggestion. Thank you.

Subject 30: "Figure 7: Why do you compare only with KAS?"

Response:

We decided to compare CO2 net biospheric flux and fossil fuel flux for European region only with maximum and minimum of CO2 from Kasprowy Wierch because the Hungarian station is under much more stronger influence of local signals and the footprint for Hegyhatsal station is distinctly smaller than in case of Kasprowy Wierch station (see Fig. 10).

Subject 31:

"Figure 8, 9, 11: Is it necessary to show all the years? You could consider showing only those important years."

Response:

We aimed to show how strong could be the differences between separate years. In our opinion, the easiest way to point these differences is to show the thermal or soil moisture anomalies for the whole period, year by year.

Subject 32: "P20 L343: Please rephrase "...Hegyhatsal station because of the larger footprint of Kasprowy Wierch and lack of" as "...Hegyhatsal station because of smaller footprint and...", i.e. be consistent with the subject"

Response:

We thank the reviewer for the suggestion. We have modified the sentence in question. It now reads: From this perspective, high-altitude mountain station Kasprowy Wierch appears to be better suited for tracing subtle, long-term changes of carbon cycle over the European continent than low-altitude Hegyhatsal station, for which a smaller footprint and stronger local signals influencing $CO_2$ mixing ratios in the PBL effectively mask larger-scale effects.

---

## Author Comment (AC3) · 24 Feb 2020

Subject 1:

"The authors did not analyse all the potential contributions that can lead to changes in the seasonal cycle amplitude, they only discuss reduced photosynthesis for an increase in the summer minimum and reduced fossil emissions for a decrease in the winter maximum. They do not even mention all the other influencing factors such as changes in ecosystem respiration, changes in the growing season length, changes in

land use, CO2 fertilisation (surely plays a role over the 24 year period) other changes in fossil fuel emission than reduced emissions from heating due to warmer winters."

Response:

We fully agree with the reviewer. In fact, our discussion looks insufficient taking into account the complexity of the CO2 balance on the continental scale. Our aim was to simplify the processes but we admit that the reduction of detailed analysis might confuse the readers. We will rewrite the text with appropriate references to each part of the budget. As our work is directed toward interpretation of measurement data, we will try to justify the influence of the particular processes using databases information, isotopic analysis of CO2, in particular 14C information.

The doubts expressed in this part of the review coincide with the question asked by the first reviewer. The answer was fully included in the commentary to the first review, subjects 1, 3 and 5.

Subject 2:

"The author also do not sufficiently support their hypothesis with other data sources. They only use results from CarbonTracker, however, integrated over the Transcom Europe region, which covers the whole geographical Europe extending eastwards to the Ural. This is a much larger region than what the footprints of the two analysed stations cover. Both the biospheric fluxes as well as the fossil fuel emissions used in Carbon-Tracker are available per grid-cell and analysing those data for the footprint regions of the stations would be more meaningful. Additional data sources (e.g. fossil fuel emissions from EDGAR, biospheric fluxes from the Global Carbon Project) for analysing the changes in emissions in Central-Eastern Europe are available, for instance, at the ICOS Carbon Portal."

Response:

The issues raised by the reviewer are mostly addressed in the responses to RC1 (see

subjects 2, 6 and 16).

Adding to our comments available there, we would like to point out that our footprint analyses were focused only on the years 2003 and 2010 and did not cover the full study period, which we have also clarified following a different comment (c.f. subject 6 below). Also, in order to properly account for the area of influence, single-trajectory analysis is not sufficient. One would need to take into account the dispersion and mixing of the air parcels, e.g. by performing ensemble calculations. Another issue is that in order to perform proposed analysis, only the areas over which simulated air parcels were actually influenced by the surface fluxes (i.e. inside the planetary boundary layer) should be counted in the footprint estimation.

To overcome these issues, our model framework would have to be significantly expanded, and it is considered as a future study. However, in the current paper, we wanted to use observations as the main source of analysis and our intention was to use Hysplit results only as a support.

Subject 3:

"Another aspect that is not discussed at all in the manuscript is how their findings of a reduced seasonal cycle amplitude relates to previous publications reporting an increased seasonal cycle amplitude in the northern hemisphere, e.g. Graven et al., 2013, and Forkel et al., 2015. Since this manuscript is rather contradicting these previous results, the changes in the seasonal cycle amplitude at the two stations subject of this paper need to be set in context with the other studies."

Response:

In Graven et al. (2013) and Forkel et al. (2016), one can find that SCA of CO2 is rising in northern ecosystems since the 60's last century. This rising trend of CO2 SCA is much stronger in the case of Barrow station, Alaska (BRW, 71oN) then in case of Mauna Loa, Hawaii (MLO, 20oN), what was shown in Fig. 1. (Graven et al., 2013).

Forkel et al. (2016) point out that "arctic and boreal regions have experienced strong warming in recent decades and a "greening" trends have been detected from satellites, indicating enhanced plant growth". But lower latitudes do not show so rapidly growing trend in CO2 SCA which was presented in both publications by comparison data form BRW and MLO (Fig. 1 in Graven et al., 2013 and Forkel et al. 2016). What is more, when we look at the SCA from MLO station since 1990 (once again Fig. 1 in both publications), the trend seems to change from positive to negative. This issue was studied for example by Yin et al. (2018). They showed that the beginning of the 21st century indicates that the fertilization process is weakening. The correlation between CO2 seasonal cycle amplitude and the temperature became negative around the year 2000 at most northern stations. It seems to confirm a limit to the "warmer spring – bigger carbon sink" mechanism. This finding highlights a dynamic temperature sensitivity of the terrestrial ecosystem to climate warming and cautions the use of current carbon-climate response to constrain future projections. This issue will be elaborated in the revised version.

References:

Graven H. D., Keeling R. F., Piper S. C., Patra P. K., Stephens B. B., Wofsy S. C., Welp L. R., Sweeney C., Tans P. P., Kelley J. J., Daube B. C., Kort E. A., Santoni G. W., Bent J. D., Enhanced Seasonal Exchange of CO2 by Northern Ecosystems Since 1960, Science 06 Sep 2013: Vol. 341, Issue 6150, pp. 1085-1089, DOI: 10.1126/science.1239207

Forkel M., Carvalhais N., Rödenbeck C., Keeling R., Heimann M., Thonicke K., Zaehle S., Reichstein M., Enhanced seasonal CO2 exchange caused by amplified plant productivity in northern ecosystems, Science 12 Feb 2016: Vol. 351, Issue 6274, pp. 696-699, DOI: 10.1126/science.aac4971

Yin, Y., Ciais, P., Chevallier, F., Li, W., Bastos, A., Piao, S., et al. (2018). Changes in the response of the Northern Hemisphere carbon uptake to temperature over the last three decades. Geophysical Research Letters, 45, 4371–4380. https://doi.org/10.1029/2018GL077316

Subject 4:

"L34-35: It is not obvious why future climate predictions from numerical climate models rely on high-quality observations of atmospheric CO2 concentrations?"

Response:

Climate predictions relay on results from numerical climate models. The direct approach to model evaluation is to compare model output with observations and analyze the resulting difference. (Flato et al., 2013) Climate models results have the uncertainty resulting mostly from poorly defined parameters of parameterized physical processes. A typical solution is to correct uncertain model parameters by analyzing the agreement of the model with available observations based on historical data. (eg. Bellprat et al., 2012). Well calibrated models can be then used for climate predictions assuming that respective emission scenarios will be met. Additional references supporting the statement will be quoted in the revised version.

References:

Bellprat, S. Kotlarski, D. Lüthi, and C. Schär, 2012, Objective calibration of regional climate models. JGR 117, doi: doi:10.1029/2012JD018262

Flato, G., J. Marotzke, B. Abiodun, P. Braconnot, S.C. Chou, W. Collins, P. Cox, F. Driouech, S. Emori, V. Eyring, C. Forest, P. Gleckler, E. Guilyardi, C. Jakob, V. Kattsov, C. Reason and M. Rummukainen, 2013: Evaluation of Climate Models. In: Climate Change 2013: The Physical Science Basis. Contribution of Working Group I to the Fifth Assessment Report of the Intergovernmental Panel on Climate Change [Stocker, T.F., D. Qin, G.-K. Plattner, M. Tignor, S.K. Allen, J. Boschung, A. Nauels, Y. Xia, V. Bex and P.M. Midgley (eds.)]. Cambridge University Press, Cambridge, United Kingdom and New York, NY, USA.

[Figure]

Subject 5:

"L 50ff: Why 'this' lack of representation, there hasn't been any mention of any lacks before."

Response:

The whole sentence has been changed as follows: "The lack of proper representation of Central-Eastern Europe in present GHG observation networks is partly compensated by Kasprowy Wierch greenhouse gas monitoring station. The station is located in the High Tatras mountain range of southern Poland, at the level of 1989 m a.s.l. This high elevation of the measuring point and lack of strong $CO_2$ sources in the direct vicinity of the station assure that the measured $CO_2$ signal is representative for Central-Eastern Europe background."

Subject 6:

"L 121-122: Do you mean by 'periods of interest' that you have calculated footprints over the whole 24 year period? I assume that this is the period of interest."

Response:

The sentence will be rephrased in the revised manuscript: "Assessment of the area of influence (footprints) for Kasprowy Wierch and Hegyhatsal stations has been carried out for the three month periods (June, July and August) in 2003 and 2010 with hourly resolution as described in the figure captions. Each frequency plot was created based on 2208 individual trajectories."

Subject 7:

"L 133: What do you mean by constant 5 cell x 5 cell weighting field?"

Response:

The description was expended to clarify the statement. It now reads:
"Following an approach previously used in Jeelani et al., 2018, aggregated gridded spatial maps of the area of influences were calculated at 0.5° x 0.5° resolution from individual back-trajectories, using the tools available in the Hysplit modeling suite. All trajectory points below 10 km altitude were included in the spatial gridding algorithm. Original footprint output was subsequently smoothed spatially using the focal function from the raster library in R software. Averaging was done using an equal-weight square matrix (5 x 5)."

Subject 8:

"L134-135: Please explain the approach and not only provide a reference, especially if it is only similar and not the same approach!"

Response:

The explanation of the footprint analysis methodology has been modified in the manuscript accordingly.

Subject 9:

"L137: How about consistency in datasets when using ERA-interim for climate extreme detection and NCEP for footprint analysis?"

Response:

NCEP data was chosen to drive the Hysplit model for several reasons: (i) it was readily available for both periods of interest (2003, 2010), (ii) it had low disk space require-ments and was already well established. While it is true that ERA-Interim (or even newer datasets like ERA5) represent the actual atmospheric state conditions better, we believe that the difference between calculated footprint areas would not be sig-nificant enough to warrant extra work necessary for the conversion of the data. At the same time, we recognize that the overall quality of the ERA-Interim dataset to be higher, which is why we used it in the - more critical to the discussion - analysis of the state of the soil during the analyzed period.

Subject 10:

"L 144: Why do you use only the uppermost soil layer? Is this the soil moisture layer which affects plant water stress? Plants usually have much deeper roots and access to soil water at deeper layers."

Response:

Thank you for pointing this out. Indeed, the 2m air temperature was used for the analysis. We modified the text to include the information as per your suggestion.

Following the comment by another reviewer, the calculation of soil humidity was also changed from using the first level only to the weighted arithmetic mean of soil humidity profile obtained by using four levels available in ERA-Interim data. The weighting function was scaled using the depth of soil layers. A detailed description was added in section 3.3.

---

## Author Comment (AC4) · 2 Mar 2020

Subject 1:

"I find the introduction unfortunately too poor in terms of framing the current study in the previous works evaluating trends in the seasonal cycle amplitude of CO2 in the Northern Hemisphere (Graven et al. 2013, Forkel et al, 2015, Piao et al. 2017, Yin et al. 2018, to name only a few)."

Response:

Thank you for this comment. This deficiency of the original MS will be corrected in the revised text. In this regard, we refer also to our response to the comments of the reviewer #1 (subjects 1 and 3) and #2 (subject 3).

Subject 2:

"Why the authors analyse the amplitude in the seasonal cycle of CO2 mixing ratios (SCA) and what is the current debate on trends and drivers of SCA - why is central Europe a region of interest and why do they claim that central Europe is poorly represented (compared to other regions in the globe, it has much higher density of stations...)."

Response:

The European continent belongs to one of the most densely populated areas of the world and is heavily industrialized. Hence, large sources of anthropogenic carbon dioxide emissions are located over a relatively small area. European climatology is complex and certainly vulnerable to the influences of climate change. The carbon emissions in the region are well cataloged, so the European continent is a very good testing ground for carbon budget models.

Station density in Europe is indeed high, but only a few of them are located in the eastern part of the continent. It was our intention to underline the importance of these eastern regions, which generally receive less attention in European carbon budgets despite the fact that they cover a substantial portion of the entire continent. Kasprowy Wierch station fills this gap, as it is exposed to westerly circulation, but also on a number of occasions, is sampling easterly air masses (c.f. Fig 10).

The text will be revised accordingly.

Subject 3:

"What is the value of long term monitoring sites used in this study – what novel aspects are brought by this study."
Response:

We appreciate this comment.

Data record from measurement sites operating coherently for a long time is relevant for the analysis of long-term changes in the carbon budget in the same way as long records of surface air temperature might be useful for the detection of climate change. In our study, we intend to emphasize the evidence for long-term changes in the regional carbon budget of Central-Eastern Europe leading to changes in SCA contrasting those reported for higher northern latitudes (e.g. Piao et al. 2017, Yin et al., 2018).

The text will be revised accordingly

We also refer to our discussion of subject 8 in this document and to our response to reviewer #1, subject 3.

Quoted references are listed at the end of this document.

Subject 4:

"Likewise, the conclusions fail to set the current findings in contrast with the previous studies, especially since they are to some extent contradictory with some studies (e.g. Graven, Forkel)."

Response:

We thank the reviewer for pointing out the literature references.

The apparent contrast between conclusions of our study and previous findings quoted by the reviewer may stem from the fact that the aforementioned studies were focused on high-latitude areas, and their results might not be directly applicable to the European continent. We discuss this more broadly in the subject 8 of this document.

The text will be revised accordingly.

Subject 5:

"Lines 231-234 Why do the authors immediately conclude that the reduction in the winter peak is due to reduction in fossil fuel emissions only? Is it not possible that biospheric processes play a role? The authors could answer this question by using FF emission data and transporting the fluxes forward to evaluate the contribution of FF to the seasonal cycle amplitude of $CO_2$ on this site. I would argue that the analysis does not settle the attribution to either anthropogenic or biospheric fluxes."

Response:

We thank the reviewer for this comment.

The issues that are raised here are commented upon in the respective responses to reviewer 1 and 2. Not to repeat the argumentation, here we will give a short answer and provide references to more detailed discussions in the other responses.

We agree that the argumentation that we have used in the manuscript should be more comprehensive, specifically concerning the relation between anthropogenic and biogenic contributions during winter (see the answer to RC01, subjects 1-5 and RC02, subjects 1-3). This expansion will be a part of the revised manuscript.

We thank the reviewer for the suggestion about estimating the seasonal cycle amplitude. This is certainly a direction we would like to take in the near future. However, in the current manuscript we wanted to limit the reliance on the modelled results, as these often have uncertainties which are not easily quantifiable (see also reply to subject 8, and the discussion in the answer to RC2, subject 2).

Subject 6:

"The authors compare the results of site-level SCA with continental averaged CTE surface fluxes, which I do not think it correct. First, the authors do not define how the European continent is defined. Secondly, the authors should only compare the fluxes from CTE that are within the site's footprint, which the authors then show in Figure 10 to be quite variable, and to not cover the full European continent. I think the appropriate

method to attribute changes in SCA to FF or Biospheric fluxes would be to transport forward the fluxes from CTE in order to calculate the resulting concentrations at HUN and KAS."

Response:

We thank the reviewer for this suggestion.

A partial answer to the raised issue has already been given in subject 5 of this comment (RC03), but a more detailed discussion addressing it can be found in our responses to reviewers #1 (RC1, subject 6) and #2 (RC2, subject 2).

Adding to the discussion raised there, in their revised manuscript we will add a precise description together with the definition of the European continent according to TRANSCOM (together with reference to Baker et al., 2006).

Subject 7:

"Thirdly, the authors use only one dataset for FF emissions and one atmospheric inversion system. However, Gaubert et al. 2019 has shown that there is large disagreement in hemispheric fluxes between different inversions systems (and smaller regions should be even more difficult to constrain), and that a large fraction of the disagreement between inversions could be attributed to the FF emission data sets used. Therefore, it would be advisable to include more atmospheric inversion datasets to obtain an uncertainty range for surface fluxes. "

Response:

We thank the reviewer for making this point and for the reference. We agree that the results of the inversion systems need to be carefully discussed. We will expand the discussion and add the reference. We would also like to refer here to our comments to RC01 (subject 7) and RC02 (subject 2).

Subject 8:

"Finally, in Fig. 7 the authors compare apples and oranges: in-situ CO2 mixing ratios in ppm/yr with continental net biospheric exchange. By doing this, the authors assume that trends in [CO2] SCA are directly linked to net annual CO2 exchange, but trends in SCA could be found even if the net annual balance would not change, for example if increased uptake in summer would be offset by increased release in autumn and winter (see Piao et al. 2008 and Figure S9 in Bastos et al. 2019 ACP). Decreasing winter amplitude could also be explained by increased photosynthesis under warmer winters (which the authors indicate in Fig. 11). As mentioned above, the only way(s) to make the attribution to different processes would be to translate CO2 surface fluxes into concentration space using an atmospheric transport model, or else to invert CO2 concentrations into fluxes, and comparing the site footprints with CTE."

Response:

We thank the reviewer for this remark.

For clarification, in Fig. 7a we are analyzing only the minimum (i.e. the departure from the trend line) of the annual concentration against the European net ecosystem exchange (NEE, estimated with CarbonTracker-EU), and we only interpret this minimum as being linked to NEE (and not the full SCA).

We agree that the relationship is not strong, which could stem from the mechanisms pointed out by the reviewer, however, on the annual basis, weaker uptake by the biosphere has to lead to less negative concentration in the summer. The mechanism is given by the reviewer as an example (i.e. that of increased summer uptake offset by increased release in autumn/winter) would indeed allow for constant annual NEE, but would also result in a stronger minimum of the summer CO2 mixing ratio, which would be visible in Fig. 7a. This, however, would have the opposite effect on the long-term trend line, which we do not observe.

Decreasing winter maximum (not 'winter amplitude'), shown in Fig. 7b indeed might partially be a result of increased winter photosynthesis during winters, however, this

effect is certainly superimposed on the reduction of fossil fuel emissions in Europe (see Fig. 6, Le Quere et al., 2018).

The relative influence of this photosynthetically increased uptake is extremely difficult to quantify, even using the state-of-the-art models. For example, in their work Piao et al. (2008) estimated that northern terrestrial ecosystems "may currently lose carbon dioxide in response to autumn warming". Such a mechanism would increase, rather than decrease, the winter CO2 peaks reported in our Fig. 7b. In turn, Yin et al. (2018) reported "negative to positive switch between fall/winter $\Delta$NBP-$\Delta$T, [which] challenged the 'warmer winter-larger carbon release' assumption", while "highlighting both the complexity of the carbon processes outside the peak growing season that requires further studies, and at the same time, the resultant uncertainty arises in our ability to project future carbon-climate feedback."

It has also to be noted that these studies, similar to Bastos et al. (2019), discuss SCANBP (seasonal amplitude of CO2 Net Biome Production), which can only be used to compare against seasonal cycle amplitude (SCA) at (i) larger spatial scales, and (ii) where the influence of regional fossil fuel emissions is negligible (as in high northern latitudes). As all the aforementioned studies were focused on high-latitude areas, their results might not be directly applicable to our study region in mid-latitude Europe.

The approach most similar to ours (with data including observations from mid-latitude Europe) was adopted by Piao et al. (2017), who reported that CO2 fertilization and climate change drove the increase in SCA for sites above 50 degrees N. However, at mid-latitude sites, land-use, oceanic and fossil-fuel fluxes, as well as trends in atmospheric transport "also contributed to the SCA trends". Flask measurements from Hegyhatsal (HUN) site were included in their analyses, but reported long-term statistical trends were smaller than reported here (0.011 ppm/year peak-to-through, -0.055 ppm/year through-to-peak), but statistically insignificant (Figure 1 in Piao et al. 2017). In the convention used by the authors, these would signify increasing annual SCA, albeit they deemed it to be statistically insignificant. The second site in Central Europe

included in that study, Baltic Sea (BAL), located at 55°21'N latitude (KAS 49°14'N, approx. 700km north; HUN 46°57'N,), yielded an opposite trend, with -0.025 ppm/year peak-to-through and +0.063 ppm/year through-to-peak, indicating a decreasing trend in SCA, albeit also statistically insignificant. Neither of those trends was exactly captured by the models used in that study, which is not unexpected due to their lower resolution and overall focus on global trends

Unfortunately, while their results are quite valuable for the ongoing discussion on the critical issue on northern-latitude SCA, it is difficult to directly compare their results with ours, as (i) the scale of their study was global and not regional, (ii) only two flask sites used in their study was located in Central Europe, and (ii) the period of the analyzed dataset was different (1980-2012).

We certainly agree that the transport and inverse models could be helpful in distinguishing between various contributions to the observed SCA signals. However, given the still significant uncertainties associated with using those, we would be hesitant to treat the models as the only way forward. This is particularly true on the regional scales.

We will expand our discussion in the revised manuscript taking into account the literature stated above. Literature: at the end of the document.

Subject 9:

"The use of statistics. The authors overstate confidence in some results that are non-significant, e.g. Lines 240-243 "as well as the growing net CO2 flux of the continental biosphere" - which is 0.03 +- 0.03, and therefore non-significantly different than zero. On the other hand, when discussing trends in Mace Head the authors state that trends are not discernible (Lines 198-199), but the value is 0.05+-0.04, which could be considered significantly increasing."

Response:

We thank the reviewer for raising this important issue, which we admit was treated too

briefly in the original version of the article.

The values of the confidence level for the given ranges will be clearly indicated in the text. The discussion of the significance of individual parameters will be conducted more explicitly and systematically. We also consider adding the p-values in the text where necessary and to graphically mark the uncertainty of the fits on the charts.

Literature:

Baker, D. F., et al. ( 2006), TransCom 3 inversion intercomparison: Impact of transport model errors on the interannual variability of regional CO2 fluxes, 1988–2003, Global Biogeochem. Cycles, 20, GB1002, doi:10.1029/2004GB002439.

Bastos, A., Ciais, P., Chevallier, F., Rödenbeck, C., Ballantyne, A. P., Maignan, F., Yin, Y., Fernández-Martínez, M., Friedlingstein, P., Peñuelas, J., Piao, S. L., Sitch, S., Smith, W. K., Wang, X., Zhu, Z., Haverd, V., Kato, E., Jain, A. K., Lienert, S., Lombardozzi, D., Nabel, J. E. M. S., Peylin, P., Poulter, B., and Zhu, D.: Contrasting effects of CO2 fertilization, land-use change and warming on seasonal amplitude of Northern Hemisphere CO2 exchange, Atmos. Chem. Phys., 19, 12361–12375, https://doi.org/10.5194/acp-19-12361-2019, 2019. Forkel, M., Carvalhais, N., Rödenbeck, C., Keeling, R., Heimann, M., Thonicke, K., Zaehle, S., and Reichstein, M.: Enhanced seasonal CO2 exchange caused by amplified plant productivity in northern ecosystems, Science, 351, 696–699, 2016.

Graven, H. D., Keeling, R. F., Piper, S. C., Patra, P. K., Stephens, B. B., Wofsy, S. C., Welp, L. R., Sweeney, C., Tans, P. P., Kelley, J. J., Daube, B. C., Kort, E. A., Santoni, G. W., and Bent, J. D.: Enhanced Seasonal Exchange of CO2 by Northern Ecosystems Since 1960, Science, 341, 1085–1089, 2013.

Piao, S., Ciais, P., Friedlingstein, P., Peylin, P., Reichstein, M., Luyssaert, S., Margolis, H., Fang, J., Barr, A., Chen, A., Grelle, A., Hollinger, D. Y., Laurila, T., Lindroth, A., Richardson, A. D., and Vesala, T.: Net carbon dioxide losses

of northern ecosystems in response to autumn warming, Nature, 451, 49–52, https://doi.org/10.1038/nature06444, 2008.

Piao, S., Nan, H., Huntingford, C., Ciais, P., Friedlingstein, P., Sitch, S., Peng, S., Ahlström, A., Canadell, J. G., Cong, N., Levis, S., Levy, P. E., Liu, L., Lomas, M. R., Mao, J., Myneni, R. B., Peylin, P., Poulter, B., Shi, X., Yin, G., Viovy, N., Wang, T., Wang, X., Zaehle, S., Zeng, N., Zeng, Z., and Chen, A.: Evidence for a weakening relationship between interannual temperature variability and northern vegetation activity, Nat. Commun., 5, 5018, https://doi.org/10.1038/ncomms6018, 2014.

Piao, S., Liu, Z., Wang, Y., Ciais, P., Yao, Y., Peng, S., Chevallier, F., Friedlingstein, P., Janssens, I. A., Peñuelas, J., and Sitch, S.: On the causes of trends in the seasonal amplitude of atmospheric CO2, Global change biology, Glob. Change Biol., 24, 608–616, 2017

Le Quéré, C., Andrew, R. M., Friedlingstein, P., Sitch, S., Hauck, J., Pongratz, J., Pickers, P. A., Korsbakken, J. I., Peters, G. P., Canadell, J. G., Arneth, A., Arora, V. K., Barbero, L., Bastos, A., Bopp, L., Chevallier, F., Chini, L. P., Ciais, P., Doney, S. C., Gkritzalis, T., Goll, D. S., Harris, I., Haverd, V., Hoffman, F. M., Hoppema, M., Houghton, R. A., Hurtt, G., Ilyina, T., Jain, A. K., Johannessen, T., Jones, C. D., Kato, E., Keeling, R. F., Goldewijk, K. K., Landschützer, P., Lefèvre, N., Lienert, S., Liu, Z., Lombardozzi, D., Metzl, N., Munro, D. R., Nabel, J. E. M. S., Nakaoka, S., Neill, C., Olsen, A., Ono, T., Patra, P., Peregon, A., Peters, W., Peylin, P., Pfeil, B., Pierrot, D., Poulter, B., Rehder, G., Resplandy, L., Robertson, E., Rocher, M., Rödenbeck, C., Schuster, U., Schwinger, J., Séférian, R., Skjelvan, I., Steinhoff, T., Sutton, A., Tans, P. P., Tian, H., Tilbrook, B., Tubiello, F. N., van der Laan-Luijkx, I. T., van der Werf, G. R., Viovy, N., Walker, A. P., Wiltshire, A. J., Wright, R., Zaehle, S., and Zheng, B.: Global Carbon Budget 2018, Earth Syst. Sci. Data, 10, 2141–2194, https://doi.org/10.5194/essd-10-2141-2018, 2018

Yin, Y., Ciais, P., Chevallier, F., Li, W., Bastos, A., Piao, S., Wang, T., and Liu, H.:

[Figure]

Changes in the response of the Northern Hemisphere carbon uptake to temperature over the last three decades, Geophys. Res. Lett., 45, 4371–4380, 2018.